# Macromolecular and electrical coupling between inner hair cells in the rodent cochlea

Philippe Jean [1,2,3,4,14], Tommi Anttonen[1,2,5,14], Susann Michanski[2,6,7], Antonio M. G. de Diego[8], Anna M. Steyer[9,10], Andreas Neef[11], David Oestreicher [12], Jana Kroll [2,4,6,7], Christos Nardis[9,10], Tina Pangršič[2,12], Wiebke Möbius [9,10], Jonathan Ashmore [8], Carolin Wichmann[2,6,7,13 ✉] & Tobias Moser [1,2,3,5,10,13 ✉]

Inner hair cells (IHCs) are the primary receptors for hearing. They are housed in the cochlea and convey sound information to the brain via synapses with the auditory nerve. IHCs have been thought to be electrically and metabolically independent from each other. We report that, upon developmental maturation, in mice 30% of the IHCs are electrochemically coupled in 'mini-syncytia'. This coupling permits transfer of fluorescently-labeled metabolites and macromolecular tracers. The membrane capacitance, $Ca^{2+}$-current, and resting current increase with the number of dye-coupled IHCs. Dual voltage-clamp experiments substantiate low resistance electrical coupling. Pharmacology and tracer permeability rule out coupling by gap junctions and purinoceptors. 3D electron microscopy indicates instead that IHCs are coupled by membrane fusion sites. Consequently, depolarization of one IHC triggers pre-synaptic $Ca^{2+}$-influx at active zones in the entire mini-syncytium. Based on our findings and modeling, we propose that IHC-mini-syncytia enhance sensitivity and reliability of cochlear sound encoding.

[1] Institute for Auditory Neuroscience and InnerEarLab, University Medical Center Göttingen, Göttingen, Germany. [2] Collaborative Research Center 889, University of Göttingen, Göttingen, Germany. [3] Auditory Neuroscience Group, Max Planck Institute of Experimental Medicine, Göttingen, Germany. [4] Göttingen Graduate School for Neurosciences and Molecular Biosciences, University of Göttingen, Göttingen, Germany. [5] Synaptic Nanophysiology Group, Max Planck Institute of Biophysical Chemistry, Göttingen, Germany. [6] Molecular Architecture of Synapses Group, Institute for Auditory Neuroscience and InnerEarLab, University Medical Center Göttingen, Göttingen, Germany. [7] Center for Biostructural Imaging of Neurodegeneration, Göttingen, Germany. [8] UCL Ear Institute and Department of Neuroscience, Physiology and Pharmacology, University College London, London, UK. [9] Electron Microscopy Core Unit, Department of Neurogenetics, Max Planck Institute of Experimental Medicine, Göttingen, Germany. [10] Center for Nanoscale Microscopy and Molecular Physiology of the Brain, University of Göttingen, Göttingen, Germany. [11] Neurophysics laboratory, Campus Institute for Dynamics of Biological Networks, University of Göttingen, Göttingen, Germany. [12] Experimental Otology Group, Institute for Auditory Neuroscience, InnerEarLab, and Department of Otolaryngology, University Medical Center Göttingen, Göttingen, Germany. [13] Multiscale Bioimaging Cluster of Excellence (MBExC), University of Göttingen, Göttingen, Germany. [14] These authors contributed equally: Philippe Jean, Tommi Anttonen. ✉email: carolin.wichmann@med.uni-goettingen.de; tmoser@gwdg.de

The well-established neuron doctrine of Ramón y Cajal states that neurons operate as independent cellular entities that communicate at specific contact points, presently known as chemical and electrical synapses[1]. Although cell–cell fusion is a normal part of the development of several non-neuronal cell types, the reticular theory of Gerlach and Golgi stating that the nervous system is a single continuous network of neural cells is considered obsolete. However, several studies indicate that neural cells are capable of fusion both under physiological and pathological conditions. For example, direct neuron–neuron fusion has been reported in C. elegans[2] but, in the mammalian nervous system, the prevalence and possible physiological significance of neuron–neuron fusion is less clear[3].

In contrast, neural communication via gap junction (GJ)-coupling is well established[4]. GJs are formed by connexins that connect the cytoplasm of two cells together and allow the propagation of electrical signals and molecules up to ~1 kDa in size[5]. Being at the core of electrical synapses, GJs are essential for synchronized spiking activity in the central nervous system[6]. In mammalian sensory systems, GJs play several important roles: GJ-coupling between retinal photoreceptors shapes visual sensitivity[7], while in the olfactory system GJs contribute to the lateral excitation of mitral cells[8].

In the cochlea, GJs between glial-like supporting cells (SCs) are critical for normal hearing, whereas inner hair cells (IHCs) are assumed to lack electrochemical coupling[9]. IHCs are the mechanosensitive receptors of hearing that transform sound born vibrations into electrical activity of the spiral ganglion neurons (SGNs). Each SGN is believed to innervate and receive input from a single IHC[10]. First steps of auditory sensory processing already occur in the cochlea: Due to its tonotopical organization, SGNs innervating IHCs of the apical coil encode low-frequency sounds, while those of the basal coil encode high-frequency sounds[11]. Here, we demonstrate that in rodents, approximately one-third of IHCs undergo homotypic macromolecular and low-resistance electrical coupling upon cochlear maturation, putatively via IHC–IHC fusion. The resulting IHC mini-syncytia form small receptive fields that pool signals and improve sensitivity and reliability of auditory coding, yet maintain its frequency resolution.

## Results

### Dye spread reveals macromolecular coupling between IHCs.

Coupling between IHCs was observed as dye-spread into neighboring IHCs (Fig. 1) in two independent laboratories combining whole-cell patch-clamp recordings with fluorescence imaging of IHCs from apical (~4–8 kHz) and mid-cochlear (~10–16 kHz) regions in 2–13-week-old mice (C57BL/6J, C57BL/6N and CD1 backgrounds; Fig. 1a). Such dye-coupling among IHCs was unlikely to have artificially resulted from cellular damage in the ex vivo preparation. It was not only found in IHCs of the excised organ of Corti (Lucifer yellow, LY, ~0.4 kDa, Fig. 1a, d, e, f), but also in a temporal bone preparation (Oregon-Green-Bapta-5N, OGB5N, ~1 kDa, Fig. 1b, c). The coupling is also metabolic as fluorescently labeled glucose was detected diffusing between IHCs (Supplementary Fig. 1a). In addition to the spread of low molecular tracers, also a larger tetramethylrhodamine (TAMRA)-conjugated CtBP2 binding peptide (~4.4 kDa) diffused between IHCs (Fig. 1g, i; Supplementary movies 1–3) as well as even large fluorophore-conjugated antibodies (~150 kDa; Fig. 1j; Supplementary Fig. 1b), indicating that the coupling mechanism, unlike GJs (size limit: ~1 kDa)[9], supports the exchange of macromolecules. Moreover, IHC macromolecular coupling is not limited to mice, as spreading of TAMRA-peptide was also observed in low-frequency (~200–450 Hz) as well as high-frequency

(~24–48 kHz) IHCs of 2–6-week-old Mongolian gerbils (Supplementary Fig. 2). Therefore, rodent IHCs are capable of forming mini-syncytia in which the receptor cells undergo diffusional exchange of molecules.

### IHC coupling is upregulated during postnatal development.

The prevalence of dye-coupling among mouse IHCs increased during postnatal IHC maturation (Fig. 2a–c). Shortly after hearing onset (P14-18)[12] only 2% of patched IHCs were dye-coupled (Fig. 2a). In mice aged P21–28, 6.6% of IHCs formed "mini-syncytia" of up to 5 IHCs (Fig. 2b). In P30–45 mice, 29% of IHCs were coupled and the distribution of IHCs coupled in the mini-syncytia (up to 7 IHCs, Fig. 2c and Supplementary Fig. 1b) significantly differed from P21–28 (Kolmogorov–Smirnov test, $p < 0.00001$). In temporal bone preparations, 33% of IHCs were coupled in P26–100 mice, with up to 9 IHCs in a mini-syncytium (Supplementary Fig. 3).

### Low resistance electrical connections between coupled IHCs.

Macromolecular coupling is accompanied by low resistance electrical connectivity between IHCs. Indeed, the maximal voltage-gated $Ca^{2+}$-influx ($I_{CaMax}$) and resting currents ($I_{rest}$) increase with the number of dye-filled IHCs (Fig. 2d, e, e', Supplementary Fig. 2b). Both parameters were significantly larger for mini-syncytia of 3 and 4 dye-coupled IHCs than for single IHCs (1 IHC: $I_{CaMax} = -161 \pm 15$ pA, $I_{rest} = -22.05 \pm 2.23$ pA, $n = 21$ (data are taken from Jean et al.[19] where similar recording conditions were used); 2 IHCs: $I_{CaMax} = -241 \pm 28$ pA, $I_{rest} = -32.15 \pm 4.39$ pA, $n = 13$; 3 IHCs: $I_{CaMax} = -356 \pm 67$ pA, $I_{rest} = -56.25 \pm 10.28$ pA, $n = 4$; 4 IHCs: $I_{CaMax} = -482 \pm 68$ pA, $I_{rest} = -50.75 \pm 6.99$ pA, $n = 4$ recordings; mean ± S.E.M, Dunn–Holland–Wolfe non-parametric multiple comparison test, $p < 0.05$ for 3 IHCs and 4 IHCs vs. 1 IHC). Interestingly, the strength of IHC coupling varied as also indicated by the variable spread of the TAMRA peptide (Supplementary Fig. 1c, c'). Strongly dye-coupled IHCs, showing a rapid dye-spread, exhibited fast monophasic $I_{Ca}$ activation kinetics. In the rare case of weakly dye-coupled IHCs, showing slow dye-spread, $I_{Ca}$ exhibited a multiphasic activation, probably due to poorer voltage-clamp of the coupled IHCs via a larger intercellular resistance (Supplementary Fig. 1d, d' and Supplementary Fig. 2b). Likewise, membrane capacitance ($C_m$) estimates, derived from capacitive currents to voltage steps, increased with the number of dye-coupled IHCs again indicating low resistance connections in the majority of the mini-syncytia. As expected, a correlation with $C_m$ was found for $I_{CaMax}$ ($r = -0.66$, $p < 0.00001$, Fig. 2e) as well as for $I_{rest}$ (Fig. 2e', $r = -0.74$, $p < 0.00001$). In a larger data set combining different $[Ca^{2+}]_e$ (1.3, 2, or 5 mM), correlations of were also found for $I_{CaMax}$ and $I_{rest}$ with $C_m$ ($I_{CaMax}$, r = $-0.66$, $p < 0.00001$, $I_{rest}$, r = $-0.64$, $p < 0.00001$, P15–51, Supplementary Fig. 1e, e'). Similar observations were made (i) from ruptured-patch recordings from IHCs of the excised apical and basal gerbil organ of Corti (P14–38, Supplementary Fig. 2), (ii) from perforated-patch recordings of mouse IHCs in the excised organ of Corti ($n = 14$ IHCs, $N_{animals} = 11$, P15–P36), and (iii) from ruptured-patch recording of IHCs when only one inner sulcus cell was removed for access of the pipette in the excised apical mouse organ of Corti ($n = 15$ IHCs; $N_{animals} = 6$, P16–P24, Supplementary Fig. 4).

In addition, we analyzed the membrane conductance as a function of the number of OGB5N-loaded IHCs in the temporal bone preparation. Using $Cs^+$ to minimize the conductance of $K^+$ channel we depolarized the patch-clamped IHCs to various levels (100 ms long depolarizations in 10 mV increments from a holding potential of $-60$ mV (Supplementary Fig. 5). The $Cs^+$

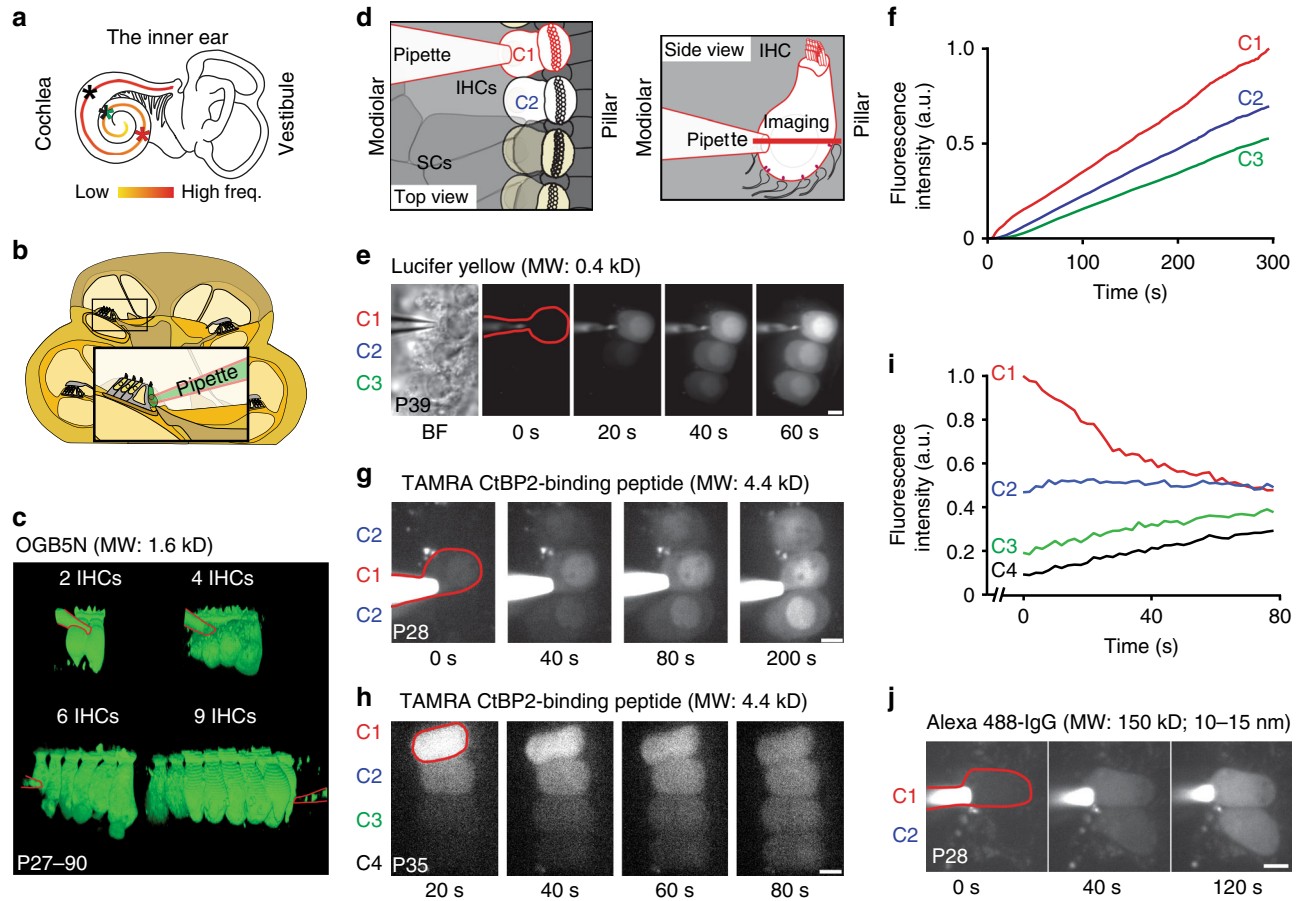

**Fig. 1 Macromolecular coupling between mature mouse IHCs. a** Illustration of the cochlea with the tonotopic map color-coded for frequency and approximate positions of patching indicated with asterisks (red = mouse temporal bone preparation; green = mouse excised organ of Corti; black = gerbil excised organ of Corti (Supplementary Fig. 2)). **b** Drawing illustrating the temporal bone preparation, where the bone covering the apical turn is removed. **c** 3D view of OGB5N fluorescence in 2,4,6, and 9 coupled IHCs, with pipette delineated in red. **d** Drawings illustrating the excised organ of Corti with SCs removed to allow patching of one IHC (C1, outlined in red in **e**, **g**, **h**, and **j**), loaded IHCs in white (C1, C2), imaging plane as red bar. **e** Time series of Lucifer Yellow loading into 3 coupled IHCs, shown in bright field (BF) on the left. **f** Normalized fluorescence intensities over time from **e**. **g** Time series of TAMRA loading into 3 coupled IHCs. **h** TAMRA-peptide diffusion between 4 coupled IHCs after retracting the patch-pipette from C1. **i** Normalized fluorescence intensities over time from **h**. **j** Time series of Alexa 488-IgG loading in 2 coupled IHCs. a.u. arbitrary units, IHC inner hair cell, MW molecular weight, P postnatal day, SC supporting cell. Scale bar = 5 µm in **e**–**j**.

currents increased with the number of dye-filled IHCs, indicating a partial summation over the currents of the coupled IHCs. We note that the recordings underestimate the true currents due to the voltage drop over the pipette series resistance to the patched IHC and due to the junctional resistance among the coupled cells. In order to quantify this result, we analyzed the membrane slope conductance ($G_m$) of IHCs at $-60$ and $+20$ mV. A significant positive correlation of $G_m$ and the number of coupled cells was observed at both potentials ($-60$ mV: $r = 0.78$; $+20$ mV: $r = 0.77$, both regression slopes significantly different from 0 at $p < 0.05$; Supplementary Fig. 6).

The above observations suggest that a coupling mechanism forms low resistance connections between IHCs. For a direct estimation of the junctional resistance ($R_J$), we performed paired voltage-clamp recordings of junctional currents ($I_j$, Fig. 3) from IHCs in the excised, apical organ of Corti of 3−5-week-old mice. The first cell, recorded in the ruptured-patch configuration was filled with LY (Fig. 3a, b, g). When recording simultaneously from the neighboring IHC, we observed bidirectional $I_J$ for dye-coupled IHCs (Fig. 3c, d), but not when LY was limited only to the loaded IHC (Fig. 3e–g). The $R_J$ between coupled IHCs ranged from 12 to

75 MΩ ($35 \pm 20$ MΩ, $n = 3$). A model of an electrical circuit equivalent of coupled IHCs with $R_J$ of 10 and 100 MΩ rendered $I_j$ values spanning the ones obtained in the paired recordings of coupled IHCs (Fig. 3h, i).

**Pharmacology rules out GJs and macropores as a mechanism.** The ability of the IHC coupling mechanism to support macromolecular and low $R_J$ electrical connectivity strongly argues against well-established mechanisms such as GJs. To further confirm this, we tested whether blockers of connexins can disrupt IHC coupling. Carbenoxolone (CBX) did not prevent IHC dye-coupling (Fig. 4a) nor changed the observed $C_m$ values (Fig. 4c). Flufenamic acid and 1-octanol, also GJ blockers[13], had no effect either (Supplementary Fig. 7). However, all three blockers reduced $I_{CaMax}$ (Fig. 4b, c and Supplementary Fig. 7), which likely reflects inhibition of $Ca^{2+}$-channels[14–16] since this effect also occurred in non-coupled IHCs (Fig. 4d, e and Supplementary Fig. 7c, d, g, h). Moreover, the P2X receptor antagonist suramin that blocks P2X7-dependent, pannexin-containing macropores[17] had no effect on $C_m$ or $I_{CaMax}$ (Supplementary Fig. 7i–l). These

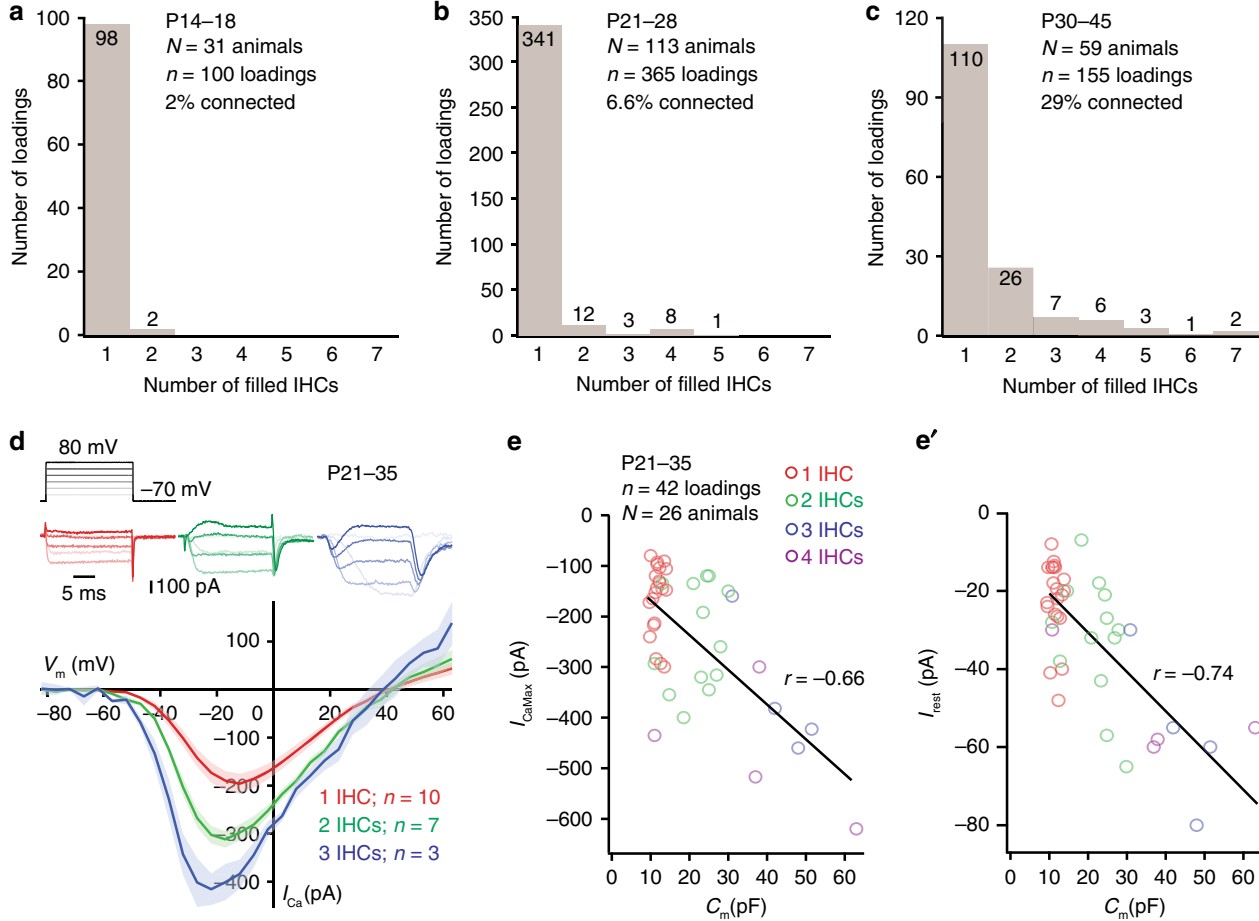

**Fig. 2 IHC coupling is age-dependent and electrical. a–c** Histograms displaying the number of dye-filled IHCs per loading at 3 different developmental stages. **d** Representative whole-cell $Ca^{2+}$-currents ($[Ca^{2+}]_e = 5$ mM) of a single IHC (red), 2 (green), and 3 (blue) coupled IHCs evoked by step depolarizations from −70 to 80 mV at P21–35 (as in **e**, **f**). Current-voltage relationships (IVs) in single and dye-coupled IHCs with mean (line) ± S.E.M. (shaded areas). Scatter plots and line fits relating $I_{CaMax}$ (**e**) and $I_{rest}$ (**e′**) currents against $C_m$ values according to the number of dye-coupled IHCs. $C_m$ membrane capacitance, $I_{CaMax}$ maximal $Ca^{2+}$-current, IHC inner hair cell, $I_{rest}$ resting current.

findings exclude GJs and P2X7-dependent macropores as mechanisms underlying IHC coupling.

**3D electron microscopy suggests IHC−IHC fusion as a coupling mechanism.** To identify a structural basis for the mechanism of IHC coupling, we employed 3D electron microscopy (focused ion beam–scanning electron microscopy (FIB-SEM): voxel size: 3 nm in x/y and 5 nm in z). We focused our imaging on the perinuclear basolateral membranes of mouse IHCs that appeared to be in contact in more mature preparations (Figs. 5, 6; Supplementary Fig. 8; Supplementary movie 4). We investigated the membranes of 3 IHCs to their respective neighboring IHCs (neighbors not being included into the IHC count) right after the onset of hearing (2 IHCs from a P15 organ of Corti and 1 IHC from a P16 organ of Corti, $n = 3$ FIB-SEM runs, $N_{animals} = 2$) and 3 IHCs after full maturation (2 IHCs from a P34 organ of Corti and 1 IHC from a P37 organ of Corti, $n = 3$ FIB-SEM runs, $N_{animals} = 2$). By performing FIB-SEM, we found that SCs usually enwrapped P15/P16 IHCs (Figs. 5a and 6d). Nonetheless, filopodia-like IHC-protrusions occasionally contacted the neighboring IHC ("filopodial contact", Figs. 5a′; 6a, d; Supplementary Table 1). This was further revealed at high resolution using electron tomography of high-pressure frozen/freeze-substituted IHCs (Fig. 6b). However, only one of these contacts appeared to harbor a putative IHC–IHC fusion site with plasma

membrane continuity and a cytoplasmic bridge (Fig. 5e, Supplementary Fig. 8a, Table 1) reminiscent of tunneling nanotubes[18].

At P34/37, SCs allowed extended membrane contacts between IHCs ("flat contacts", Fig. 5c; Fig. 6a, f; Supplementary Table 1; Supplementary movie 4). Some of these contacts featured putative fusion sites with cytoplasmic bridges that occasionally contained ribosomes (Fig. 5d, e; Supplementary Fig. 8b; Supplementary movie 5). The bridges measured, excluding one outlier, $27.9 \pm 6.5$ nm along the z-axis and $69.3 \pm 20.6$ nm in the xy-plane ($n = 7$ sites in P34/37 IHCs, Table 1). The low number and size of the detected putative fusion sites indicate that this process appears to be self-limiting. In contrast to IHCs, outer hair cells (OHCs) are well separated by SCs. Using serial block face-scanning electron microscopy we did not find direct contacts between P22 OHCs in 34 cases studied (Supplementary Fig. 9; Supplementary movie 6).

**Collective synaptic activity in IHC mini-syncytia.** In order to elucidate physiological consequences of IHC coupling, we performed spinning-disk confocal imaging of $Ca^{2+}$-signals at the individual presynaptic active zones (AZs) in IHC-mini-syncytia. To do so we loaded the mini-syncytium with the low-affinity $Ca^{2+}$ indicator Fluo-4FF, the TAMRA-conjugated CtBP2-binding peptide, and EGTA via the patch-clamped IHC. Under these conditions the $Ca^{2+}$ indicator fluorescence serves as a proxy of synaptic $Ca^{2+}$ influx[19–21]. Depolarization of the patch-clamped

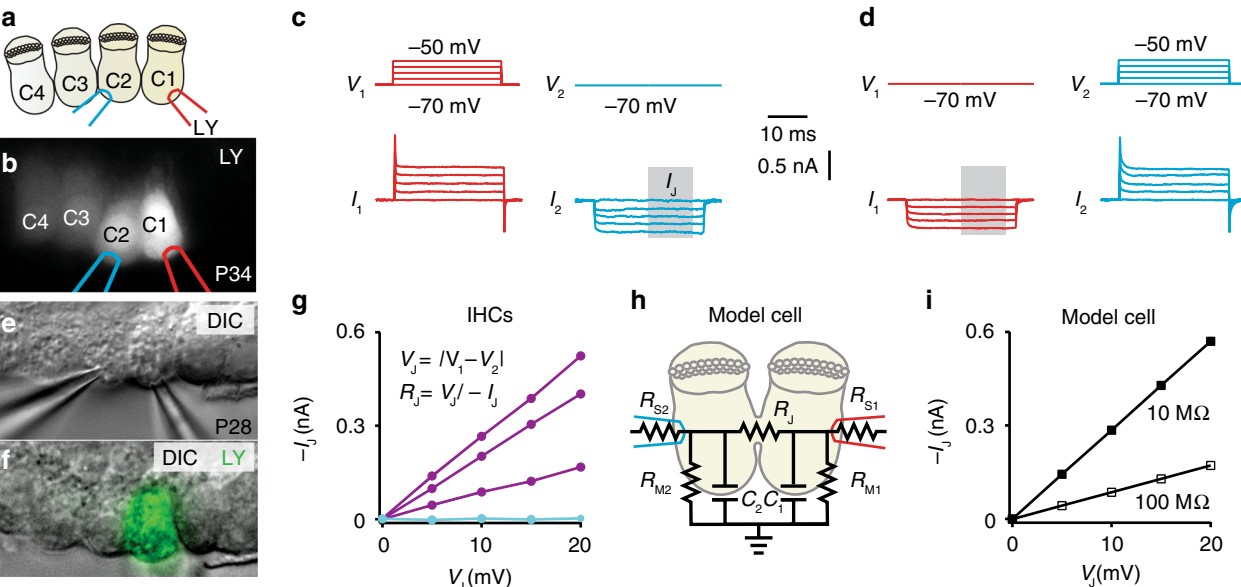

**Fig. 3 Dual whole-cell voltage-clamp recordings from dye-coupled IHCs reveal bidirectional low resistance electrical coupling. a, b** After observing IHC-to-IHC diffusion of LY from the loaded IHC (C1), the second neighboring coupled IHC (C2) was patched with a LY-free pipette (left). **c, d** Exemplary dual recording of coupled IHCs (3 recordings in total) demonstrates how depolarizations of one of the IHCs from holding potential ($V_H$, −70 mV) produces junctional currents ($I_J$) in the second IHC held at $V_H$. **e, f** In the absence of dye coupling, no $I_J$ was observed (cyan in **g**). **g** The steep linear $I_J$-$V_J$ curve indicates a large, ohmic conductance between the dye-coupled IHCs (purple, $n = 3$). Negligible conductance is observed in non-coupled IHCs (cyan, $n = 3$). **h** An electrical circuit equivalent of coupled IHCs (cell capacitance, $C_1 = C_2 = 10$ pF; junctional resistance, $R_J = 10$ MΩ (filled box in **i**) or 100 MΩ (empty box in **i**); Cell membrane resistance, $R_{M1} = R_{M2} = 1$ GΩ; Series resistance, $R_{S1} = R_{S2} = 10$ MΩ) (**i**) produces $I_J$ values that frame those of coupled IHCs ($n = 3$, compare with **g**). $V_J$, junctional voltage difference, $I_J$ junctional current, LY lucifer yellow.

IHC evoked $Ca^{2+}$-signals at the TAMRA-marked AZs within the entire IHC-mini-syncytium (Fig. 7, Supplementary Fig. 9 for additional examples and Supplementary movie 7), likely activating dozens of postsynaptic SGNs. This indicates that SGNs innervating a certain IHC can be activated via the receptor potential shared in the IHC mini-syncytium. We did not find differences of the synaptic $Ca^{2+}$-signals between the patched IHC (C1) and its coupled neighbor(s) (C2, the number of spots analyzed from further distant coupled cells did not permit statistical comparisons). Neither their maximal amplitude ($\Delta F_{max}/F_0$: 1.48 ± 0.25, $n = 37$ spots in C1 of 8 mini-syncytia, $N_{animals} = 8$ vs. 1.88 ± 0.28, $n = 43$ spots in C2 of 8 mini-syncytia, $N_{animals} = 8$; $p = 0.31$, Mann–Whitney–Wilcoxon test) nor their voltage-dependence of activation ($V_h$: −23.66 ± 2.24 mV, $n = 17$ spots in C1 in 7 mini-syncytia, $N_{animals} = 7$ vs. −23.79 ± 1.59 mV, $n = 30$ spots in C2 in 8 mini-syncytia, $N_{animals} = 8$; $p = 0.96$, Student $t$-test; note that because of signal–to–noise requirements the analysis of voltage-dependence could only be performed on a subset of synapses) were different.

**Coupling improves signal-to-noise ratio of sound encoding.** Sensory coding, including cochlear sound encoding, is inherently noisy. This noise originates from fluctuations at the transduction and synaptic encoding stages that reflects the stochasticity of ion channels gating and neurotransmitter release. Electrical coupling of sensory cells is expected to attenuate the noise of the receptor potential by averaging the transduction currents of the individual cells. For neighboring IHCs that face a common mechanical input resulting from the sound-driven traveling wave, such electrical intercellular pooling would further reduce fluctuations in mechanotransducer noise at low sound levels.

To test this idea, we used the Neuron simulation environment to model sound encoding at the IHC synapse and signal detection in the cochlear nucleus (CN), where coincident input from several SGNs drives temporally precise spiking of bushy cells (e.g. ref. [22]). In the model (see Supplementary Note 1 for details), a deterministic sinusoidal stimulus governs the stochastic gating of mechanotransducer channels in IHCs and the ensuing current drives excursions of the membrane voltage. This receptor potential, in turn, controls $Ca^{2+}$ channel gating (Fig. 8a–e, Supplementary Fig. 11) and long (≥ 2 ms) single $Ca^{2+}$ channel openings are considered to trigger neurotransmitter release events. Pruned by refractoriness, these release events translate to action potentials (APs) in SGNs (Fig. 8f, g, Supplementary Fig. 11).

The model indicates that signal encoding by presynaptic $Ca^{2+}$ channel openings and postsynaptic AP rates in the cochlea is subtly improved when based on an input coming from SGNs driven by AZs of a mini-syncytium rather than AZs from non-coupled IHCs (see Supplementary Note 1 for details). For example, $Ca^{2+}$ channel gating tended to be more representative of the mechanical input in coupled than in non-coupled IHCs (Pearson correlation coefficient: 0.88 for coupled IHCs vs. 0.82 for non-coupled IHCs). In addition, coincidence signal detection in the CN seemed also slightly improved (2–3%, Supplementary Fig. 12; Supplementary Note 1).

**Frequency resolution of sound encoding seems unaltered.** While IHC coupling appears to be beneficial for sound signal encoding, the fusion of the mechanoreceptive fields of each individual IHC into a larger shared field may lead to poorer sound frequency resolution. To evaluate the effects of IHC coupling on frequency resolution, we estimate that the broadest observed mini-syncytium (9 IHCs) covers approximately 1.29% of the length of the cochlea or 0.06 octaves (see Supplementary Fig. 13 and Supplementary Note 2 for details). This is less than the physiological estimates of frequency resolution at the level of the basilar membrane (0.17 octaves at the 50–56 kHz region[23])

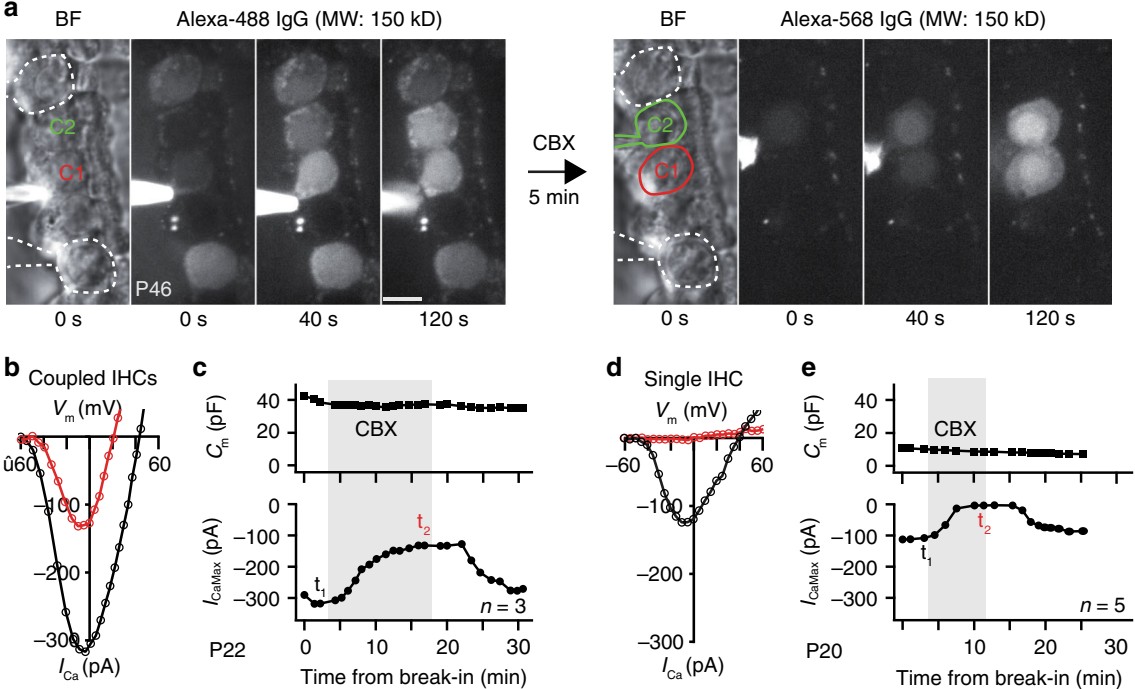

**Fig. 4 Molecular and electrical IHC coupling is not prevented by the GJ blocker CBX. a** The left time series of green fluorescence shows how Alexa-488 conjugated antibodies diffuse from the targeted IHC (delineated in red, C1) to the coupled IHC (delineated in green, C2). Two of the surrounding uncoupled IHCs (highlighted by white lines) have previously been loaded with the antibodies. After retracting the pipette, CBX (250 μM) was added to the bath solution five minutes prior to the loading of Alexa-568 conjugated antibodies via patching C2. The right time series of red fluorescence (same length scale as in **a**) shows that the spread of Alexa-568 conjugated antibodies to C1 is not blocked by CBX. Surrounding SCs, especially inner pillar cells, exhibit auto-fluorescence. **b–e** $Ca^{2+}$-current voltage (IV) curves were obtained by depolarizing IHCs from the holding voltage (−60 mV) with 5 mV steps before and during bath application of CBX. The presence of CBX is marked with gray in the membrane capacitance ($C_m$)-time and maximal $I_{Ca}$ ($I_{CaMax}$)-time plots. The recording time points of the presented IV curves are marked with $t_1$ (for black curve) and $t_2$ (for red curve) in the $I_{CaMax}$-time plots. The experiments were stopped when the patched IHC was lost. CBX inhibits $I_{Ca}$ of coupled IHCs without changing $C_m$, as seen in with single IHCs. CBX carbenoxolone, $V_m$ membrane voltage. Scale bar: **a** = 5 μm.

and auditory nerve fibers (0.37 octaves at the 12 kHz region[24]) of the mouse cochlea. For the human cochlea, we estimate that a single IHC covers approximately 3 cents (1 cent is 1% of a semitone, one semitone is 1/12th of an octave) on the tonotopic map of the mid-cochlear quarter of a turn. By assuming that the observed coupling of rodent IHCs also occurs similarly in humans and the whole cochlea is hypothetically formed by tiles of IHC mini-syncytia with the average size of 3 IHCs, the best possible frequency resolution would be 9 cents. This is remarkably close to the lowest psychophysical estimates of frequency resolution (7 cents[25]). Therefore, the low prevalence and small size of mini-syncytia avoids trading off frequency resolution against detection sensitivity.

## Discussion

The prevalent view on cochlear sound encoding pictures IHCs as single receptor units that provide sampling of the mechanical stimulus with the highest frequency resolution. In addition, each SGN is believed to receive sensory information from a single IHC. Our results challenge the above views and indicate IHC mini-syncytia to represent auditory receptor fields of the cochlea next to those of uncoupled IHCs. Such patchy information mixing downstream of the mechanical input offers distinct advantages to the system and appears compatible with the frequency resolution of physiological sound encoding. Unlike photoreceptors, IHCs employ GJ-independent coupling. IHC coupling likely arises from cell-to-cell fusion and results in exchange of macromolecules and low resistance electrical connectivity. IHC coupling might be a more general feature of cochlear organization, at least in rodents,

as we found it in mice and gerbils, regardless of the tonotopic position.

Based on combining studies of dye diffusion, electrophysiology, and 3D electron microscopy we propose macromolecular IHC coupling via self-limiting cell-to-cell fusion. This seems highly unorthodox at first but it is supported by the unusual coupling properties that are at odds with well-established GJ mechanisms of cell coupling. Firstly, IHC coupling allows the spread of molecules at least as large as fluorophore-conjugated antibodies (~150 kDa), exceeding the size limit of GJs (~1 kDa[9]) by two orders of magnitude. Secondly, it offers low resistance electrical connectivity with an average junctional resistance of ~35 MΩ. Thirdly, inhibitors of connexins as well as suramin, a blocker of P2X7-dependent, pannexin-containing macropores, failed to prevent dye diffusion and to electrically isolate coupled IHCs. Finally, FIB-SEM provided evidence consistent with IHC–IHC fusion sites at extended contacts formed by their basolateral membranes. The size of the fusion sites, around 50 nm in width and length, seems compatible with the passage of macromalecules or even small organelles. It is also consistent with single-junction resistances of tens to a few hundred MΩ, so that between one and five fusion sites in parallel would result in the experimentally observed junctional resistances of ~35 MΩ. Testing whether fusion sites, indeed, mediate the observed coupling and elucidating their molecular regulation remains another important task for future studies including in IHCs.

What are the molecular mechanisms of the putative IHC–IHC membrane fusion and how could this be limited in space and frequency? We find that the IHC coupling is slowly established

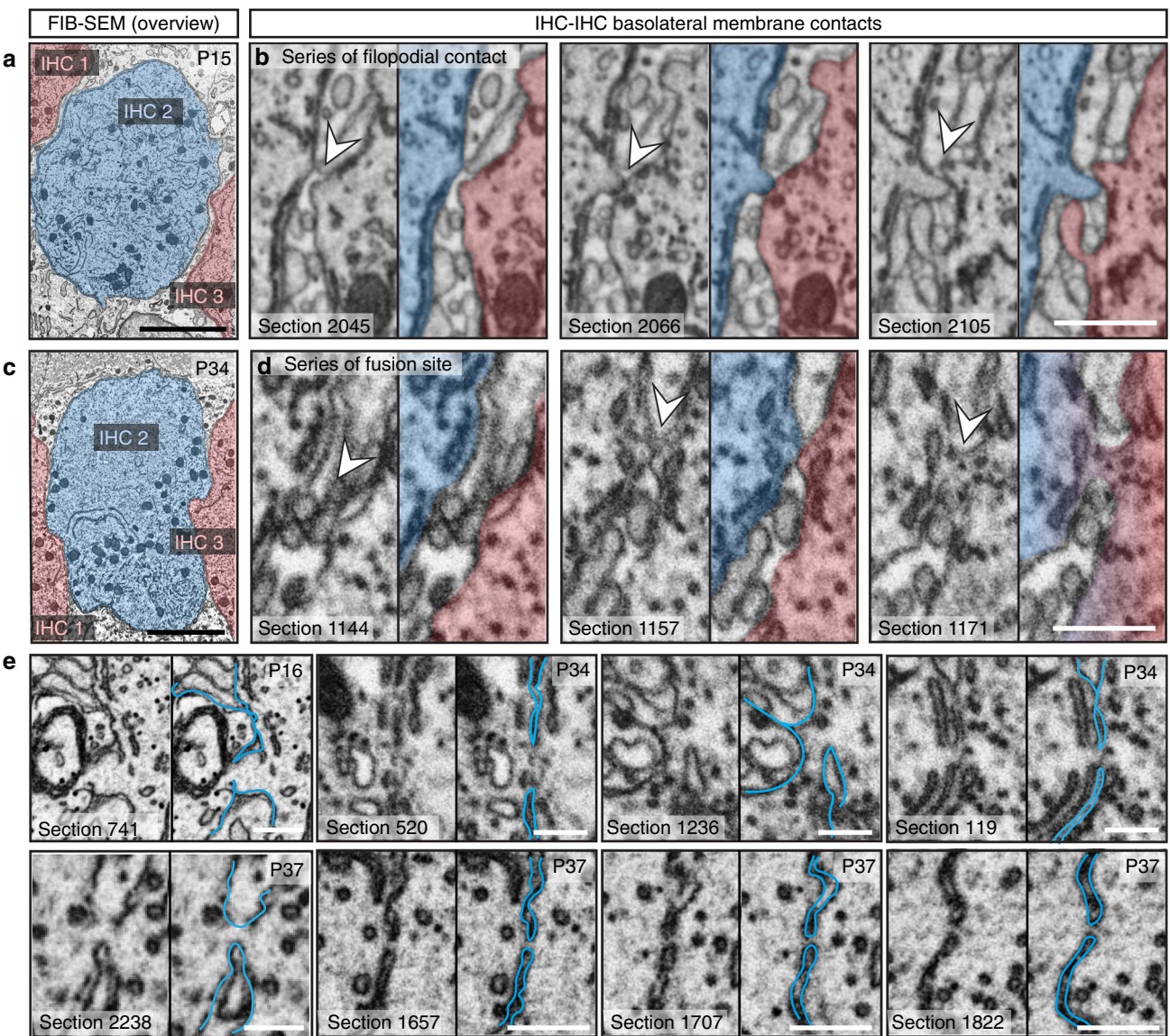

**Fig. 5 3D electron microscopy indicates IHC–IHC fusion as a putative basis for IHC coupling. a–e** FIB-SEM datasets of basolateral membrane contacts between IHCs. **a** P15 IHCs are tightly wrapped around by SCs and are separated by a distance ranging from 100 to 648 nm (first section below the nucleus). **c** P34 IHCs share prominent flat membrane contacts (blue–red contacts). For each FIB-run we numbered the IHCs, whereby IHC 2 is the full IHC in the centre of the region of interest (highlighted in blue), making contacts to the neighboring IHCs (shown in red). **b–e** Representative micrographs of IHC–IHC contacts. P15 IHCs frequently show filopodia reaching neighboring IHCs (**b**). **d** At P34, some flat membrane contacts appear to include fusion sites with membrane continuity and cytoplasmic bridge (white arrowheads). **e** Depicted are the observed putative IHC fusion sites in the P16 and the P34/37 animals. $n$ FIB-SEM run (P34/37) = 3, $N_{animals}$ = 2; $n$ FIB-SEM run (P15/16) = 3, $N_{animals}$ = 2; 2 independent embeddings for each age group. FIB-SEM focused ion beam-scanning electron microscopy, IHC inner hair cell, SC supporting cells. Scale bar: **a**, **c** = 5 μm; **b**, **d** = 500 nm; **e** = 200 nm.

after the initiation of hearing function in mice, reaching the prevalence of 30% P30–P45 IHCs forming mini-syncytia consisting of 3 IHCs in average, up to 9 IHCs. We speculate that IHC coupling is a developmentally regulated process, driven by IHC-specific fusogenic machinery, which can take action more efficiently once SCs partially retract from between IHCs. While the identification of the critical molecular components of the machinery regulating such fusion is beyond the scope of the present study, future work could consider manipulating the function of myosin X[26] expressed in IHCs[27]. In addition, other molecules involved in the formation of tunneling membrane nanotubes might also operate in putative IHC–IHC membrane fusion.

As cell fusion has been reported to occur in neural cells upon cellular damage[3] and our data have been obtained from ex vivo preparations of the cochlea, artefactual IHC fusion needs to be considered. Several lines of evidence argue against tissue damage as the cause of IHC coupling. Cell injury related to tissue handling would be expected to also induce fusion of IHCs to SCs. As we never observed dye spread from IHC to SCs, the molecular coupling is specific to IHCs (homotypic coupling). IHC recordings revealing $Ca^{2+}$ current amplitudes worth multiples of the single IHC $Ca^{2+}$ current were also observed 20 min post-mortem in the ruptured-patch clamp configuration without removing the contacting SCs (Supplementary Fig. 3). Therefore, IHC coupling was not simply induced by potential cellular stress associated with rupturing the cell membrane and removing adjacent structures. Resting currents, a sensitive parameter reflecting cell health, were not significantly different between non-coupled IHCs before and from P28 in our

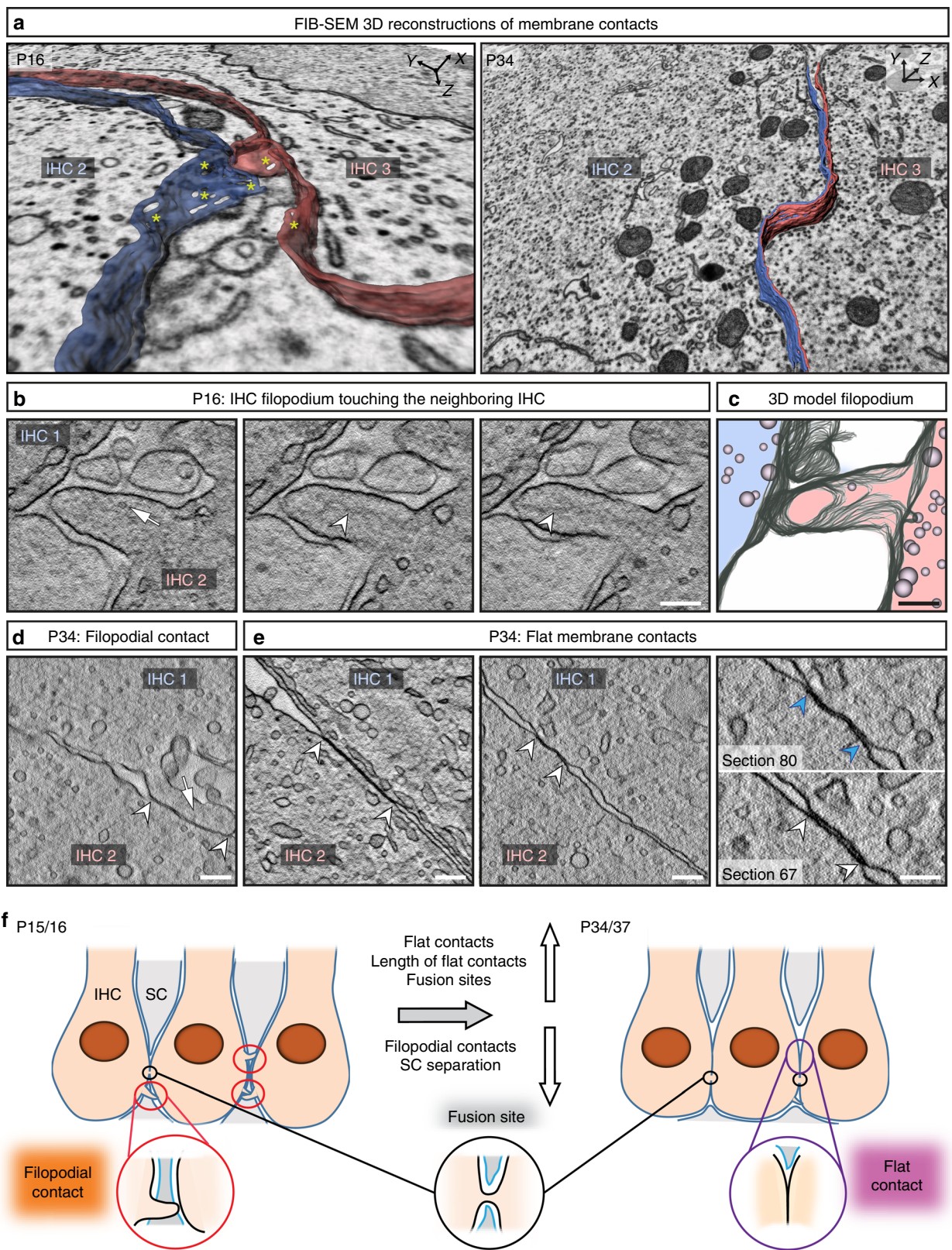

**a** FIB-SEM 3D reconstructions of membrane contacts

**b** P16: IHC filopodium touching the neighboring IHC

**c** 3D model filopodium

**d** P34: Filopodial contact

**e** P34: Flat membrane contacts

Section 80

Section 67

**f** P15/16                                   P34/37

Flat contacts
Length of flat contacts
Fusion sites

Filopodial contacts
SC separation

IHC   SC

Fusion site

Filopodial contact

Flat contact

recordings from the excised gerbil organ of Corti ($p = 0.47$; Supplementary Fig. 14). This argues that increased probability of IHC coupling in the more mature preparation does not reflect compromised IHC health in the older preparations. In a temporal bone preparation, which require little dissection of the cochlea and maintains its integrity in a near-physiological state,

IHC coupling was observed 15 min post-mortem. Moreover, during up to 1-h long patch-clamp recordings we never observed an IHC spontaneously becoming coupled to a previously non-coupled neighboring IHC. This indicates that ex vivo conditions as artificial as a long-lasting ruptured-patch recording do not readily induce IHC coupling. Direct

**Fig. 6 FIB-SEM and electron tomography indicate close membrane contact sites between neighboring IHCs. a** 3D model of a filopodial contact in immature (P16) IHCs and a flat membrane contact in mature (P34) IHCs. Yellow asterisks highlight non-traceable membrane regions. **b** Example tomographic virtual sections from a high-pressure frozen/freeze-substituted (HPF/FS) P16 sample showing a filopodium reaching from one IHC to a neighboring IHC (10 virtual sections in z between two pictures, each section distance of 2.616 nm), arrow points to a vesicle, arrowheads to cytoskeletal filaments within filopodium. **c** 3D model of the filopodium. **d** Example tomographic virtual section from HPF/FS P34 IHCs depicting a filopodium (arrow) between two adjacent cells. The arrowheads indicate the membrane contact. **e** Further HPF/FS examples showing two virtual sections of flat IHC–IHC membrane contacts with different extensions in P34. Right panel: Two virtual sections from the same tomogram as in the middle panel. The membranes come very close and cannot be distinguished anymore in virtual section 80 (blue arrowheads), whereas two separate membranes are visible in section 67 (white arrowheads). Independent HPF/FS for electron tomography of each age group = 1; n tomograms (P16) = 1, $N_{animals}$ = 1, n tomograms (P34) = 12, $N_{animals}$ = 2. **f** Schematic illustrating the development of IHC membrane contacts based on the results from FIB-SEM and electron tomography: red circles: filopodial contacts, black circles: fusion sites, purple circles: purple flat membrane contacts. In immature organs of Corti, SCs enwrap the IHCs, while mature IHCs are forming extended flat cell-to-cell membrane contacts. FIB-SEM focused ion beam-scanning electron microscopy, IHC inner hair cell, SC supporting cell. Scale bar: **b**–**e** = 200 nm.

**Table 1 Summary of the contact site quantification of the FIB-SEM datasets.**

| Age group | Total n of contacts | Filopodial contact | | Flat contact | | Fusion site | | |
|---|---|---|---|---|---|---|---|---|
| | | Av. length, z | n | Av. length, z | n | Av. length, z | Av. length, xy | n |
| P15/16 | 36 | 114 ± 17.4 nm | 32 | 42.3 ± 3.70 nm | 3 | 35.0 nm | 136 nm | 1 |
| P34/37 | 23 | 198 ± 7.50 nm | 2 | 1.69 ± 0.57 μm | 14 | 27.9 ± 6.53 nm | 69.3 ± 20.6 nm | 7 |

Summary of the total number of contacts, of filopodial contacts, flat contacts, and putative fusion sites as well as of their average length in z-plane. Note, that we excluded one putative fusion site depicted in Fig. 5e, from the statistics of the P34 data as an outlier, as the z-measurements of 535 nm clearly exceeds 1.5 times the interquartile range above the third quartile (75th percentile). Further, the maximum extent of the cytoplasmic bridge of putative fusion sites was measured in the xy-plane using 3dmod, as indicated in Supplementary Table 1, and averaged. For the age-group of P15/16 only 1 example for a fusion site was found and the respecting values are listed in the Table. Data are presented as mean ± S.E.M. P15/16: n = 3 FIB-runs (2 FIB-runs from P15, 1 FIB-run from P16) each with 1 full IHC and the respective contacts to neighboring IHCs, $N_{animals}$ = 2. P34/37: n = 3 FIB-runs (2 FIB-runs from P34, 1 FIB-run from P37) each with 1 full IHC and the respective contacts to neighboring IHCs, $N_{animals}$ = 2. Per age group 2 independent embeddings were performed.

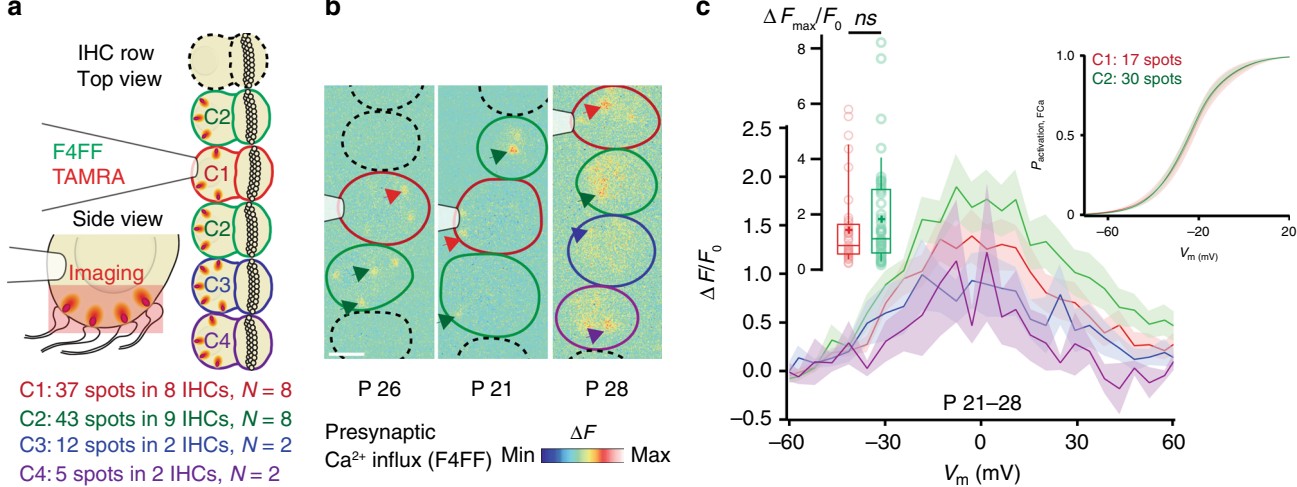

**Fig. 7 Collective synaptic activity in IHC mini-syncytia. a** Coupled IHCs (delineated in color) were patch-clamped and loaded with the calcium indicator Fluo-4FF (F4FF) and TAMRA-fluorescent CtBP2-binding peptide. C1 (red) is the patched IHC, C2 (green) are coupled and adjacent to C1, C3 (blue) is coupled to C2 and C4 (violet). The red box shows the imaging plane. **b** $Ca^{2+}$-signals co-localizing with the presynaptic ribbons are observable. Example $\Delta F/F_0$ pictures from a single plane of the $Ca^{2+}$-increments (black arrows) of 2 (left), 3 (middle), and 4 (right) coupled IHCs. Non-coupled IHCs are delineated in dashed lines. **c** The graph displays the fluorescence versus voltage-relationship ($\Delta F/F_0$ versus depolarization level), showing presynaptic activity in the coupled IHCs. Left inset show individual values and percentiles (10th, 25th, 50th, 75th and 90th) of $\Delta F_{max}/F_0$ of synapses in C1 (37 spots in 8 IHCs, N = 8) and C2 (43 spots in 9 IHCs, N = 8). Means are indicated as crosses. Right inset displays averaged fractional activation curves of a subset of the spots for which the $\Delta F/F_0$ could be fitted faithfully (C1: 17 spots in 7 IHCs, N = 7; C2: 30 spots in 8 IHCs, N = 8). Mean (line) ± S.E.M. (shaded areas) are displayed. IHC inner hair cell. Scale bar: **b** = 5 μm.

demonstration that IHC coupling exists in vivo is a challenging task, as cell-level recording and imaging methods typically require the opening of the cochlea. Therefore, new approaches will be required to probe IHC coupling in vivo.

What then are the physiological implications of IHC mini-syncytia interspersed between non-coupled IHCs along the tonotopic axis of the organ of Corti? Coupling into mini-syncytia confers considerable advantages for sharing of metabolites and

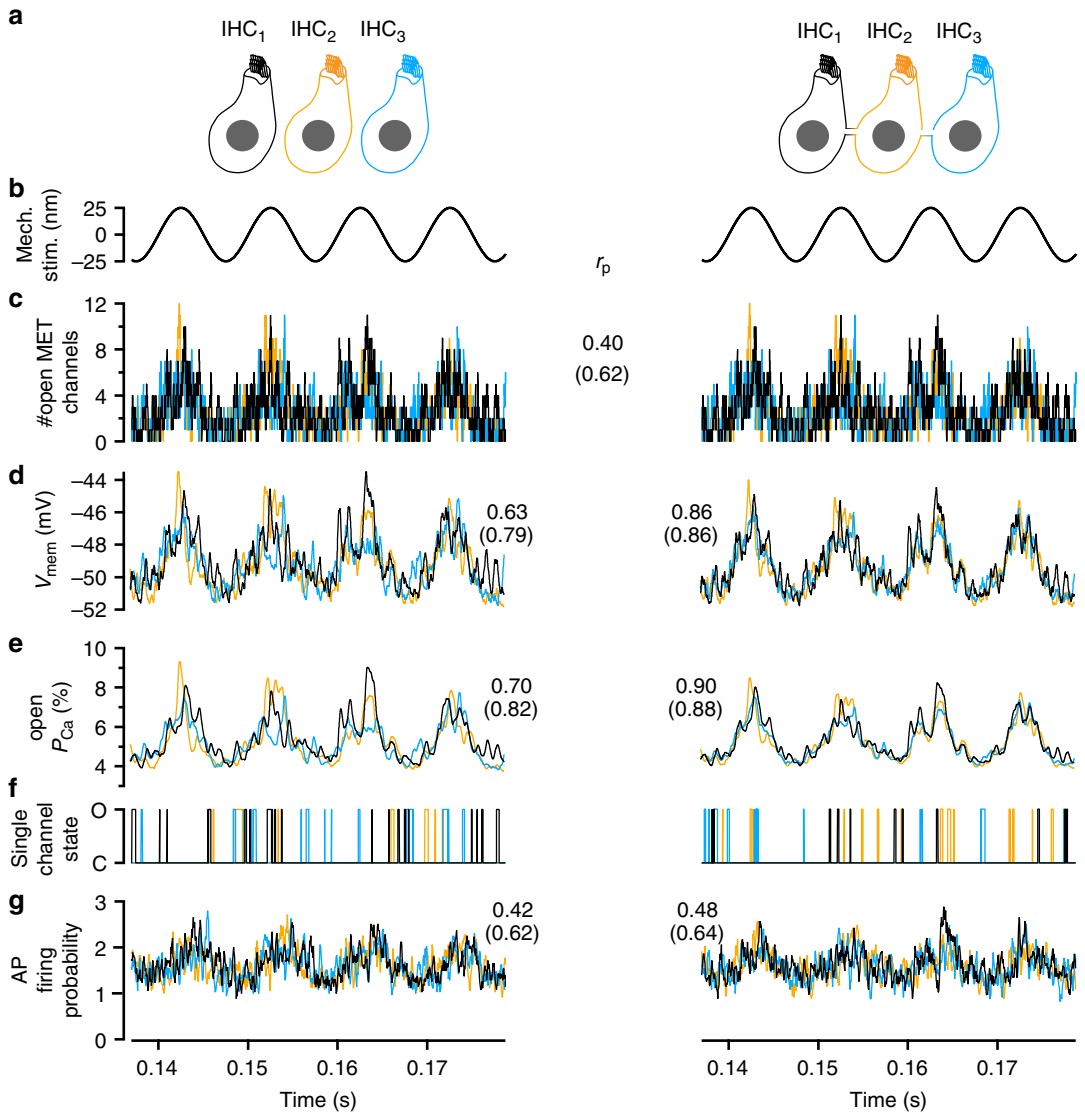

**Fig. 8 Electrical IHC coupling coordinates Ca²⁺ channel activity.** The signaling steps of IHCs and their synapses, simulated in the Neuron modeling environment for three non-coupled IHCs (left) and three coupled IHCs with a junctional resistance of 25 MΩ (right). **a** Simplified representation of IHCs, color-coded for the simulation results presented below. **b** Depiction of the sinusoidal mechanical stimulus (identical for all three cells). **c** Realization of the stochastic gating of mechanotransducer channels (specific for each cell, identical for coupled and uncoupled cells). **d** Membrane voltage variations of the three IHCs. They differ from each other because the mechanotransduction currents differ, but that difference is mitigated by electric coupling (right). **e** Illustration of the average open probability, as would result from very many realizations of Ca²⁺ channel gating given the voltage trajectories above. **f** Different conductance states (O open, C closed) of one channel in each IHC. **g** Modeling stimulus-secretion-coupling and refractoriness for 35,000 independent realizations of (**f**), an approximate probability to observe an AP in a SGN during a 200 μs window is calculated. Numbers in the central column are Pearson correlation coefficients $r_p$, averages of correlations between pairs of traces from two different IHCs. In parenthesis are the average correlations between the IHC parameter and the sinusoidal stimulus. For those correlations, time lags of 0.3 ms for the voltage and 0.5 ms for the Ca²⁺ channels are compensated for. AP action potential, IHC inner hair cell.

potentially even organelles between cells. As adult mammals cannot regenerate functional IHCs, IHCs must survive and remain functional for a lifetime—otherwise sensorineural hearing loss develops. By being able to share metabolites and a joint Ca²⁺-buffering capacity, coupled IHCs can potentially withstand higher levels of cellular stress than a single IHC alone. On the other hand, initiation of cell death processes in one of the coupled IHCs puts all of them in danger if the macromolecular coupling cannot be terminated. Future studies should test the implications of IHC coupling to cell survival.

Beyond the advantage for IHCs, our study indicates that coupling also benefits the sound encoding by SGNs. Patch-clamp combined with Ca²⁺ imaging demonstrated a collective response

of the presynaptic AZs within the IHC mini-syncytium, wherein the maximal amplitude and voltage-dependence of the AZs of the patched and the neighboring cells were not different. While the activation time course of the collective Ca²⁺ current suggests minimal if any temporal delays in recruiting the AZs, scrutinizing the synchrony of synaptic transmission in future experiments remains an important task. Regardless of the degree of synchrony, for slowly fluctuating stimuli the shared receptor potential would govern the release at all AZs. Then, a single SGN encodes information on mechanical stimuli sensed and averaged across the IHCs of the mini-syncytium, even though it receives transmission only from one AZ of one IHC. A coupled IHC with stereociliar damage could still be driven by the shared receptor

potential of the mini-syncytium, such that the SGNs postsynaptic to this IHC can contribute to sound encoding provided a fit neighbor.

Comparing sound encoding in conditions with and without the coupling described here for rodent IHCs will be instrumental to experimentally probe the advantage predicted by the computational model (Fig. 8). Ideally, such studies could be performed in the same species, e.g. in mice with genetic disruption of IHC coupling and will help to validate the notion that the low prevalence and small size of mini-syncytia avoids degradation of the frequency resolution of sound coding. Beyond this, the possible existence in other mammals remains to be studied. For example, it will be interesting to see whether IHCs inside the "acoustic fovea" of certain mole rats[28], owls[29], and echo-locating bats[30] are coupled; proposing an additional cellular candidate mechanism for this enigmatic phenomenon.

## Methods

**Animals**. Postnatal day (P) 14–100 mice from C57BL/6J, C57BL/6N, and CD1 background and Mongolian gerbils (P16–38) were used. The animals were either housed in local animal facilities in Göttingen (three different facilities) and London or imported from Charles River Germany straight into the experiment in Göttingen (to avoid potential effects of keeping mice in the local facility on the observed IHC-coupling). Real-time PCR assays in cochleae of mice bred in one of the Göttingen animal facilities did not indicate the presence of Herpes simplex Virus 1/2 or Varizella Zoster Virus, proposed to induce neuron–neuron fusion (1), arguing against virus-mediated IHC-coupling. All experiments complied with national animal care guidelines and were approved by the University of Göttingen Board for Animal Welfare and the Animal Welfare Office of the State of Lower Saxony, or by the UCL Biological Services Animal Ethics Committee.

**Tissue dissection and solution for patch-clamp experiments**. Excised organ of Corti: mouse and gerbil organs of Corti were isolated in HEPES Hank´s solution, which contained (in mM): 5.36 KCl (6781.1, Carl Roth, Germany), 141.7 NaCl (3957.1, Carl Roth, Germany), 10 HEPES (H3375, 006K5424, Sigma, Germany), 0.5 MgSO$_4$.7H$_2$O (63140-500G-F, #SBZF2890V, Sigma, Germany), 1 MgCl$_2$ (Fluka, Germany), 1 mg/ml D-glucose (G8270-1KG, #SZBF3280V, Sigma, Germany), and 0.5 mg/ml L-glutamine (G3126-100G, #SLBS8600, Sigma, Germany) (pH 7.2, ~300 mosmol/l). The tissue was transferred to a recording chamber and fixed in place with nylon threads. All experiments were conducted at room temperature (20–25 °C). The glutamate-based intracellular solution for rupture-patch/Ca$^{2+}$-imaging in mouse IHCs contained (in mM): 111 Cs-glutamate (C1251-500G, #SLBM4966V, Sigma, Germany), 1 MgCl$_2$, 1 CaCl$_2$ (Fluka, Germany), 10 EGTA (E3889, #SLBP2806V, Sigma), 13 TEA-Cl (T2265, #BCBF2894V, Sigma), 20 HEPES (H3375-500G, #SLBS2995V, Sigma, Germany), 4 Mg-ATP (A9187-100MG, Sigma), 0.3 Na-GTP (A2383-, G, Sigma, Germany) and 1 L-Glutathione (G4251-5G, #SLBD3561V, Sigma, Germany) (pH 7.3, ~290 mosmol/l). The gluconate-based intracellular solution for ruptured-patch used for the gerbil IHCs contained (in mM): 130 Cs-gluconate (HB4822, Hello Bio), 1 MgCl$_2$, 0.8 EGTA, 0.4 BAPTA tetra-Cs salt (B1212 Lot. 567406, Invitrogen), 10 TEA-Cl, 10 4-AP (A78403-25G, Sigma), 10 HEPES, 4 Mg-ATP, 0.3 Na-GTP (pH 7.2, ~290 mosmol/l).

The extracellular solution contained the following (in mM): 2.8 KCl (6781.1, Carl Roth, Germany), 102 NaCl (111; in the experiments with gerbil IHCs), 10 HEPES, 1 CsCl (C3011-50G, #SLBP4992V, Sigma, Germany), 1 MgCl$_2$, 5 CaCl$_2$ (1.3; in the experiments with gerbil IHCs), 35 TEA-Cl, and 2 mg/ml D-Glucose (pH 7.2, ~300 mosmol/l), and was perfused at 1 ml/min.

The experiments testing the effects of GJ blockers (see below) contained:

For ruptured-patch either the glutamate-based intracellular solution (see above) or (in mM): 125 Cs-methanesulfonate (C1426-25, MKBT4630V, Sigma, Germany), 10 TEA-Cl, 10 4-AP, 10 HEPES, 2 Mg-ATP, 0.3 Na-GTP, 0.5 EGTA, 0.5 BAPTA-tetra-Cs salt, and 1 MgCl$_2$ (pH 7.2. ~280 mosmol/l).

For perforated-patch, the intracellular solution contained (in mM): 130 Cs-methanesulfonate, 10 TEA-Cl, 10 4-AP, 10 HEPES, and 1 MgCl$_2$ (pH 7.2, ~290 mosmol/l). Amphotericin B (250 µg/ml Calbiochem, La Jolla, CA) was added from a 1000× stock in DMSO (final DMSO concentration not exceeding 0.1%) during sonication. Pipettes were tip-dipped in amphotericin-free pipette solution for 20–40 s and then brought to the IHC within less than 1 min to avoid accidental spill of amphotericin, which blocks seal formation. Perforation of the patch membrane usually occurred within 5 min and recordings were started when the series resistance was lower than 30 MΩ. A sudden drop of series resistance, which would be indicative of a spontaneous rupturing of the patch membrane was not observed.

The extracellular solution for perforated-patch recordings contained (in mM): 112 NaCl, 30 TEA-Cl, 5.8 KCl, 2 Na-Pyruvate (8793.2, 061168024, Carl Roth, Germany), 0.7 NaH$_2$PO$_4$-H$_2$O (1.063460500, A0536046421, Merck, Germany), 1 CsCl, 1 MgCl$_2$, 1.3 or 2.0 CaCl$_2$, and 2 mg/ml D-glucose (pH 7.2, ~305 mosmol/l).

Temporal bone preparation: The mouse temporal bone containing the semicircular canals and the cochlea was removed and placed in extracellular solution containing (mM): 140 NaCl, 1.3 CaCl$_2$, 2.1 MgCl$_2$, 5.4 KCl, and 10 HEPES. It was attached with cyanoacrylate glue to the base of the recording chamber so that the modiolus was vertical. The osmolarity of the extracellular solution was adjusted to 320 mosmol/l with glucose and the pH was adjusted to 7.4 with NaOH. The bone covering the apical turn of the organ of Corti and Reissner's membrane was removed using fine forceps to gain access to the scala media, exposing a small opening to observe the structures below between the 10 and 30% positions along the partition from the apex. After picking away the bone, the tectorial membrane covering the organ of Corti was gently removed using a glass micro-pipette while observing the organ under a microscope. The total procedure took no longer than 15 min.

We do not consider the above-described differences in ionic composition of intra- and extracellular solution of relevance for the phenomenon of IHC-coupling that was observed in every set of solutions used. The differences merely reflect the collaborative nature of the study and finding the IHC-coupling regardless of the different conditions speaks to the robustness of the finding.

**Patch-clamp of IHCs**. Excised organ of Corti: Patch-pipettes (3–7 MΩ in bath) were pulled (P87 puller, Sutter Instruments, USA) from borosilicate glass capillaries (GB150F-8P and GB150-8P Science Products, Germany) and coated with Sylgard. IHCs were visualized with fixed stage microscopes (Examiner or Axioskop 2 FS Plus, both Zeiss, Germany or BX50WI and BX51, Olympus, Japan) equipped with 63 × /1.0 DIC (Zeiss) or 60 × /0.9 DIC (Olympus) objectives and motorized micromanipulators (MP-285, Sutter Instruments or Luigs & Neumann SM-5).

EPC-9 or -10 USB double amplifiers controlled by Pulse or Patchmaster (HEKA Elektronik, Germany) were used for the measurements. IHCs were held at −87 mV (−84 and −60 mV with gluconate- and methanesulfonate-based solutions, respectively). All voltages were corrected for liquid junction potential offline (−17 mV with glutamate-, −14 mV with gluconate- and −10 mV with methanesulfonate-based solutions) and voltage-drops across the series resistance (R$_s$).

To achieve the whole-cell voltage-clamp configuration, GΩ seals were obtained by manoeuvring the patch-pipette to the IHC basolateral membrane. Access for the patching pipette was gained by removing SCs surrounding the IHC on their modiolar (neural) side except for dual voltage-clamp recordings of coupled IHCs that were performed from the pillar (abneural) side. A subset of studies was done with a minimally invasive approach, which involves patching IHCs through a small opening produced by detaching a single inner sulcus cell (2). Pipettes used to remove SCs were pulled (L/M-3P-A, List-Medical, Germany) from sodalime glass capillaries (article code 1412227, Hilgenberg, Germany).

Current-voltage relationships (IVs) were obtained by applying 10 or 20 ms depolarizing step pulses of increasing voltage from the holding voltage in 5 or 10 mV steps. IVs were used to determine the $I_{CaMax}$, which depends on the developmental stage of the IHC, the external Ca$^{2+}$ concentrations, and the health of the IHC. Measurements of resting $C_m$ were obtained with the capacitance compensation "auto C-slow" feature of Pulse/Patchmaster software. This was preceded by the compensation of the fast capacitive currents with the "auto C-fast" function. The typical $C_m$ of a single mouse IHC is ~10 pF. The $C_m$ of a single gerbil IHC is ~14 pF. Of note, the used circuitry could not fully compensate the slow capacitive currents associated with large IHC mini-syncytia (more than two coupled IHCs), rendering the $C_m$ value imprecise in these occasions.

Temporal bone preparation: The bone was stuck to a 35 mm dish and the cells observed either by white light using transmitted from the substage through a hole in the overlying bony capsule. Judicious placing also gave access to the recording pipette. Where necessary the overlying tectorial membrane was removed by suction.

**Live-fluorescence imaging**. Excised organ of Corti: The fluorescent markers used were Lucifer yellow CH (dilithium salt, L0259-25MG, Lot# MKBS3318V, Sigma-Aldrich, Germany), F4FF (Fluo-4FF penta-K$^+$ salt, 0.8 mM, Life Tech.), TAMRA-conjugated CtBP2/RIBEYE-binding dimer peptide (10 µM, Biosynthan, Berlin, Germany), Glucose-Udp-Fluorescein conjugate 488 (3% dilution in intracellular solution, SMB00285-100UG, Sigma, Germany), Alexa Fluor 488 goat anti-rabbit IgG and 568 anti-rabbit IgG (3% dilution in intracellular solution, A11008 and A11011 respectively, Life technologies, Germany). To visualize the Ca$^{2+}$-hotspots and the presynaptic ribbons, the Ca$^{2+}$-indicator Fluo-4FF (Fluo-4FF penta-K$^+$ salt, 0.8 mM, Life Tech.) and the TAMRA-conjugated CtBP2/RIBEYE-binding dimer peptide were added to the intracellular solution. In addition to binding the ribbon-component RIBEYE, the peptide binds to nuclear C-terminal binding protein 2 (3). Ca$^{2+}$ imaging was performed with a spinning disk confocal scanner (CSU22, Yokogawa, Germany) mounted on an upright microscope (Axio Examiner, Zeiss, Germany) with 63×, 1.0 NA objective (W Plan-Apochromat, Zeiss, Germany). Images were acquired by a CMOS camera (Neo, Andor). Ca$^{2+}$-indicator Fluo-4FF and TAMRA-conjugated peptide were excited by diode-pumped solid-state lasers with 491 nm and 561 nm wavelength, respectively (Cobolt AB, Sweden). The spinning disk was set to 2000 rpm to synchronize with the 10 ms acquisition time of the camera. Using a piezo positioner for the objective (Piezosystem, Jena, Germany), a scan of the entire cell was performed 4 min after breaking into the cell, taking sections each 0.5 µm at an exposure time of 0.5 s in the red (TAMRA-peptide)

channel from the bottom to the top of the cell. In order to study the voltage-dependence of $Ca^{2+}$ indicator fluorescence increments at the presynaptic active zones, the confocal scans were acquired every 0.5 μm from the bottom to the top ribbon. $Ca^{2+}$ currents were evoked by applying a voltage ramp stimulus from −87 to + 63 mV during 150 ms (1 mV/ms) in each focal plane. Simultaneously, measurements of green Fluo-4FF fluorescence were made with a frame rate of 100 Hz. In order to overcome the limitations of the frame rate and increase the voltage resolution of the fluorescent signal acquired, the voltage ramp protocol was applied twice, once shifted by 5 ms such that for any given frame during the second ramp the voltage was shifted by 5 mV compared with the first stimulus. Alternating planes were acquired to avoid photobleaching encountered with the consecutive plane acquisition.

For gerbil experiments, TAMRA-conjugated peptide or Oregon-Green-Bapta-5N (OGB-5N) was excited by a narrow band light source (Polychrome IV, Till Photonics, Germany) at a wavelength of 568 or 488 nm, respectively. Fluorescence images were obtained by a CCD camera (VGA, Till Photonics, operated by Till-Vision software) directly after breaking in to the cell using a frame rate of 0.5 Hz and an exposure time of 1500 ms.

Temporal bone preparations: Cells were imaged using a Zeiss LSM510 upright microscope using a 63 × NA 1.0 UV/Vis water-immersion objective. OGB-5N (100 μM, Lifesciences, UK) in the recording pipette was excited at 935 nm by the beam from a Chameleon Ti-Sapphire laser (Coherent, UK). The fluorescence was collected using a 505−700 band pass filter with the LSM510 operated in multi-photon mode. The Z-scans were performed either taking 0.5 μm steps (2, 4, 6 coupled cells) or 1.25 μm steps (9 coupled cells). Image reconstruction was accomplished by using Image J.

**Applying GJ-blockers and suramin to mouse organs of Corti**. After the whole-cell patch-clamp configuration on an IHC was established and several IVs were recorded, the bath solution was rapidly replaced (10 ml solution change within 2 minutes) with an extracellular solution containing one of the following compounds: Carbenoxolone disodium salt (250 μM, CBX, C-4790, Lot 22K1147, Sigma, Germany), Flufenamic acid (100 μM, FFA, F9005, #BCBV0673, Sigma, Germany), 1-Octanol (1 mM, OCT, 297887, Lot# STBG1046V, Sigma, Germany), Suramin sodium salt (100 μM, S2671-100MG, 070M1539V, Sigma, Germany). The FFA stock solution was dimethyl sulfoxide-based (DMSO, D8148, #064K00671, Sigma, Germany). DMSO (0,01% v/v) alone did not have observable effects on $C_m$ or $I_{CaMax}$ and no obvious rundown of $I_{CaMax}$ was observed during the ~30 min recording of IHCs without the GJ blockers.

**High-pressure freezing and freeze substitution**. High-pressure freezing and freeze substitution were performed as follows. Organs of Corti from P16 (C57BL/6 N; $N_{animals}$ = 1) and P34 (C57BL/6 J; $N_{animals}$ = 2) wild-type animals were dissected in HEPES Hank´s solution. Organs of Corti from P34 mice were placed directly in aluminium specimen carriers with 0.2 mm cavity (type A, Leica Microsystems, Wetzlar, Germany). In a subset of experiments (P16) they were initially transferred into the following solution (in mM) 65.4 KCl, 79.7 NaCl, 2 CaCl₂, 1 MgCl₂, 0.5 MgSO₄, 10 HEPES, 3.4 L-glutamine, and 6.9 D-glucose, pH 7.4 for 5 min and then placed onto aluminium specimen carriers. A second specimen carrier (0.1 mm cavity, type B, Leica Microsystems) was dipped in hexadecene and placed onto the first specimen carrier with the cavity showing upwards. Subsequently, all samples were frozen with an EM HPM100 (Leica Microsystems, Germany) and transferred into liquid nitrogen for storage. For freeze substitution, organs of Corti were transferred into an EM AFS2 (Leica Microsystems, Germany) precooled to −90 °C. 4-day incubation in 0.1% tannic acid in acetone was followed by three washing steps (each for 1 h in acetone) and application of 2% (w/v) osmium tetroxide in acetone at −90 °C. The temperature was increased from 90 °C to −20 °C (5 °C/h), stayed at −20 °C for 17 h and was further increased from −20 °C to + 4 °C (10 °C/h). Osmium tetroxide was removed and the organs of Corti were washed 3 × 1 h with acetone while further slowly increasing the temperature. At room temperature, the samples were infiltrated with epoxy resin (acetone/ epoxy resin 1:1 (v/v) for 2 h, 100% epoxy resin overnight (Agar Scientific 100 kit, Plano, Germany), placed into embedding molds and polymerized for 48 h at 70 °C.

**Electron tomography**. For electron tomography of the perinuclear and basolateral region of IHCs, 250 nm sections of the embedded organs of Corti were obtained using an ultramicrotome (UC7, Leica Microsystems, Germany) with a 35° DiATOME diamond knife (Science Services GmbH, Germany) and applied to formvar-coated copper 75-mesh grids (Plano, Germany). Sections were post-stained with uranyl acetate replacement (Science Services, Munich, Germany) for 40 min and lead citrate for 1 min following standard protocols, before 10 nm gold beads (British Bio Cell, UK) were applied to both sides of the grid as fiducial markers. Single tilt series from −60° to +60° (increment 1°) were acquired with a JEM2100 (JEOL, Freising, Germany) and Serial-EM (6) software at 200 kV and 5000× magnification. For tomogram generation, the IMOD package etomo (7) was used, models were generated with 3dmod (bio3d.colorado.edu/imod/).

**Enhanced en bloc staining**. The apical turn of organs of Corti (C57BL/6 J: $N_{animals}$ = 1: P15, $N_{animals}$ = 1: P16, $N_{animals}$ = 1: P34 and $N_{animals}$ = 1: P37) were dissected in HEPES Hank´s solution and fixed with 4% paraformaldehyde and 0.5% glutaraldehyde in PBS (pH 7.4) for 1 h and 2% glutaraldehyde in 0.1 M sodium cacodylate buffer (pH 7.2) overnight on ice. Afterward, samples were incubated with 1.5% potassium ferrocyanide and 4% osmium tetroxide (v/v in 0.1 M sodium cacodylate buffer) for 1 h on ice and were briefly washed in distilled water. Subsequently, specimens were treated with thiocarbohydrazide solution for 20 min followed by brief washing steps with cacodylate buffer. An additional incubation with 2% osmium tetroxide (v/v in 0.1 M sodium cacodylate buffer) was performed and the samples were briefly washed before placing them into 2.5% uranyl acetate (v/v in distilled water) overnight at dark at room temperature. After another round of washing steps, specimens were contrasted with Reynold's lead citrate for 30 min at 60 °C, washed, dehydrated in increasing ethanol concentrations, infiltrated and embedded in Durcupan (Sigma Aldrich, Germany) to polymerize for 48 h at 60 °C.

**Focused ion beam scanning electron microscopy**. The datasets of P15 ($N_{animals}$ = 1; $n$ = 2 FIB-runs) and P34 ($N_{animals}$ = 1; $n$ = 2 FIB-runs) organs of Corti were collected in a previous study (9) and reanalyzed for the present study with the focus on the membranes of the IHCs and the supporting cells, which was not the focus of the previous study. In addition, new FIB-SEM runs were performed for the ages P16 ($N_{animals}$ = 1; $n$ = 1 FIB-run) and P37 ($N_{animals}$ = 1; $n$ = 1 FIB-run) that were embedded for this study. With the use of the sputter coating machine EM ACE600 (Leica Microsystems, Germany) at 35 mA current, samples were coated with an 8 nm platinum layer. Subsequently, samples were transferred into the Crossbeam 540 focused ion beam scanning electron microscope (Carl Zeiss Microscopy GmbH, Germany) and positioned at an angle of 54°. A 300 nm carbon/platinum layer was deposited on top of the region of interest and Atlas 3D (Atlas 5.1, Fibics, Canada) software was utilized to collect the 3D data. Specimens were exposed to the ion beam driven with a 15/30 nA current while a 7 nA current was applied to polish the surface. Images were acquired at 1.5 kV using the ESB detector (450/1100 V ESB grid, pixel size x/y 3 nm) in a continuous mill and acquire mode using 1.5 nA for the milling aperture (z-step 5 nm). Finally, data were inverted, cropped, and aligned using the Plugin "Linear Stack Alignment with SIFT" in Fiji. The smoothing function was applied (3 × 3) followed by local contrast enhancement using a CLAHE plugin in Fiji, followed by a binning in x/y (10). Regions of the 3D data stack were cropped and loaded into microscopy image browser (MIB) to segment the membranes using the membrane tracer tool (11). The model was imported into Imaris for 3D visualization and finally producing a movie.

**Serial block-face electron microscopy (SBEM)**. Two datasets of P22 organs of Corti collected with SBEM in a previous study (12) were reanalyzed to search for possible physical contacts between outer hair cells. The datasets were segmented and 3D visualized with Imaris.

**Data analysis**. Live-imaging and IHC-patch-clamp data were analyzed using Fiji (Image J 1.8) and custom programs in Igor Pro 6.3 (Wavemetrics, Portland, OR, USA). For analysis of time series of fluorescent marker diffusion (Fig. 1 f, i, Supplementary Fig. 1c′), the fluorescence intensities of each frame were background (extracellular fluorescence) subtracted and normalized to the maximal fluorescence of the patched cell. The time series pictures were equally adjusted within each panel for display purposes (Fig. 1e–j, Supplementary Fig. 1a, b and c, Supplementary Fig. 2a). For analysis of IV-curves, the evoked $Ca^{2+}$ current was averaged from 3 to 8 ms, 5 to 10 ms, or 13 to 18 ms) after the start of the depolarization. ΔF images were generated by subtracting the fluorescence intensity inside the cell at the resting state ($F_0$, an average of 10 frames) from the one at the depolarized state (an average of 6 frames during voltage ramp protocol). ΔF was calculated as the average of a 3 × 3 pixel square placed in the region showing the greatest intensity increase within the fluorescence hotspot. Maximal ΔF ($ΔF_{max}$) was the average of 5 neighboring ΔF values at the peak of $Ca^{2+}$ influx obtained during the voltage ramp protocol. Only fluorescent increments presenting a $ΔF_{max}$ greater than the average of the fluorescence intensity plus 2 standard deviations at the resting potential were defined as synaptic $Ca^{2+}$ signals and considered for further analysis. Due to their variance, analysis of their voltage dependence was performed on fits to the raw FV traces using the following function:

$$F(V) = F_0 + \frac{f_v \cdot (V_r - V)}{1 + e^{\frac{(V_h - V)}{k}}} \tag{1}$$

where $V$ stands for the voltage command. The fitting parameters were determined in Igor Pro software, and their initial guess resulted from the estimations of $F_0$, the signal at rest, $V_h$ for the voltage value of half-maximal activation and $k$ for the voltage sensitivity obtained from a sigmoid fitting. The slope factor $f_v$ was obtained by linear fitting of the FV-trace in the range of 3 to 23 mV, where the decrease of fluorescence at positive voltages results from the declining driving force despite full activation of the $Ca^{2+}$ channels. The resulting fitting trace was forced to reach the reversal potential $V_r$, obtained from the corresponding whole-cell $Ca^{2+}$ currents. The FV fit was then divided by the $f_v$ line extended to all the corresponding voltages, to estimate the assumed fluorescence intensity of every voltage in the full activation condition, generating the fractional activation curves. The fractional activation curves were then fitted by the Boltzmann function to obtain the voltage for half activation ($V_h$) and slope-factor (k).

The $R_I$ values obtained from dual patch-clamp experiments of coupled IHCs were compensated for series and cell input resistances as described in (12). For the analysis of the FIB-SEM data sets all stacks were inspected independently by two of the authors demarking the virtual sections where direct membrane-to-membrane contact sites: flat or filopodial along with putative fusion sites were located. We operationally defined fusion sites as a continuous, traceable membrane between both IHCs forming a cytoplasmic bridge.

**Statistical analysis**. The data were analyzed using Igor Pro 6 (Wavemetrics Portland, OR, USA). Averages were expressed as mean ± standard error of the mean (S.E.M.). For every dataset the number of replicates ($n$) and animals ($N$) were indicated. In order to compare two samples, data sets were tested for normality of distribution (Jarque–Bera test) and equality of variances (F-test), followed by two-tailed unpaired Student's $t$-test, or, when data were not normally distributed and/or variance was unequal between samples, the unpaired two-tailed Mann–Whitney–Wilcoxon test was used. For multiple comparisons, statistical significance was considered as $p < 0.05$, and was calculated by using Kruskal–Wallis (K–W) test followed by non-parametric multiple comparisons test (NPMC, Dunn-Holland-Wolfe test) for non-normally distributed and unequally sized datasets. In case other tests were used this is stated in the results or supplement sections. Pearson correlation coefficient ($r$) was used to measure the linear correlation between variables. See Table S2 for a summary of the statistical tests used.

**Reporting summary**. Further information on experimental design is available in the Nature Research Reporting Summary linked to this paper.

## Data availability

Data supporting the findings of this manuscript are available from the corresponding authors upon reasonable request. A reporting summary for this article is available as a Supplementary Information file.

## Code availability

Custom routines are available at http://www.innerearlab.uni-goettingen.de/materials.html.

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

## Acknowledgements

We thank M.M. Picher and S. Krinner for contributions in an early phase of the study, U. Pirvola for providing the SBEM datasets, and A.J. Goldak, S. Gerke, and C. Senger-Freitag for expert technical assistance. We would like to thank V. Rankovic for contributions to the study. We also would like to thank H. Eiffert and R. Lugert for performing the real time PCR assays for viruses, and C. Grabner, E. Neher, D. Richter, S. Rizzoli, N. Strenzke, H. Wagner, and R. Weiler for feedback on the manuscript. This work was supported by grants of the German Research Foundation through the collaborative research center 889 (projects A02 to T.M., A07 to C.W. and B09 to T.P.), the Leibniz program (to T.M.), Niedersächsisches Vorab (T.M.) and EXC 2067/1- 390729940 (MBExC to T.M.). The FIB-SEM and the position of the FIB-SEM operator (A.M.S.) were funded by the Cluster of Excellence and Deutsche Forschungsgemeinschaft (DFG) Research Center Nanoscale Microscopy and Molecular Physiology of the Brain (CNMPB). AM.G.D. and J.A. were supported by the Wellcome Trust.

## Author contributions

P.J., T.A., C.W., and T.M. designed the study. P.J. performed the dye loading experiments (Fig. 1; Supplementary Fig. 1, Supplementary movies 1–3) with help of T.A. P.J. performed Ca$^{2+}$ imaging (Fig. 7, Supplementary Fig. 10, Supplementary movie 7). P.J. and T.A. performed the electrophysiological recordings (Fig. 2). T.A. studied the effects of gap junction blockers and suramin with help of P.J. (Fig. 4 and Supplementary Fig. 4). T.A. performed and analyzed the dual patch clamp experiments of IHCs (Fig. 3), least invasive patch-clamp experiments (Supplementary Fig. 4) and analyzed the SBEM datasets (Supplementary Fig. 9; Supplementary movie 6). S.M. produced/analyzed and A.M.S. and T.A. helped in analysis of FIB-SEM datasets (Figs. 5, 6 and Supplementary Fig. 8). J.K., S.M., and C.W. produced and analyzed electron tomographic datasets (Fig. 6). A.M.G.D and J.A performed temporal bone preparation and dye loading (Fig. 1, Supplementary Fig. 3), analyzed the cesium currents (Supplementary Fig. 5) and membrane conductances (Supplementary Fig. 6). A.M.S. performed and supervised FIB scans with support of W.M. and further processed and analyzed the 3D data. A.N. performed mathematical modeling of signal encoding and coincidence detection. D.O. performed electrophysiological experiments on gerbil IHCs under supervision of T.P (Supplementary Figs. 2 and 14). C.N. segmented the FIB-SEM data sets and prepared the

supplementary movies 4 and 5 with the help of A.M.S. P.J., T.A., C.W., and T.M. wrote the manuscript. All authors reviewed the manuscript.

## Competing interests

The authors declare no competing interests.
