## [Peer Review File · Nature Communications]

Reviewers' comments first round:

Reviewer #1 (Remarks to the Author):

This exceptionally interesting manuscript describes the findings that may significantly change current understanding of the cell-to-cell communications in the mammalian organ of Corti. According to the wisdom of the field, the major mechanosensory receptors of the inner ear – the inner hair cells (IHCs) – send their signals to the auditory nerve fibers separately and do not communicate to each other. However, the authors of this study discovered that, at the latest stages of postnatal development, IHCs form functional mini-syncytia interspersed between non-coupled IHCs along the tonotopic axis of the organ of Corti. The existence of these three-to-nine cells mini-syncytia was confirmed by dye diffusion, patch-clamp recordings, calcium imaging, and 3D electron microscopy. The data are of high-quality and quite convincing. Even though the functional significance of the IHC coupling remains purely speculative, the study may have a profound impact on our understanding of the mechanisms of sound encoding in the mammalian cochlea. I was very excited by this story and have only few minor comments.

As the authors correctly pointed out, the major concern is a potential artefactual IHC fusion due to cellular damage in an ex vivo preparation. The authors outlined in the Discussion several reasons to believe that such an artefact is unlikely. However, I would be more comfortable if the authors would show, on a separate figure, the data of a crucial experiment that they described as, "IHC recordings revealing Ca²⁺ current amplitudes worth multiples of the single IHC Ca²⁺ current were also observed ... without removing the contacting supporting cells."

Specific comments:

Figure 2f: Please clarify which values the data points (open circles and red crosses) represent. I assume I_{CaMax} and I_{rest}, correspondingly. If so, why I_{CaMax} data are presented separately for ruptured and perforated-patch experiments, while I_{rest} data are combined? Exactly which data were used for linear fits? There are three color-coded groups and three linear fits could be expected.

Figure S2c: Please clarify which values the data points (gray open circles and red crosses) represent. I assume I_{CaMax} and I_{rest}, correspondingly. What is the group with n=49? Is that a total number of the cells? Why is the number of cells in panel c not matching to the number of cells in panel b?

Legends to figure 3, 4, and 7 are missing the ages of the preparations.

Results, last paragraph: "...we estimate that a single IHC covers approximately 3 cents on the tonotopic map of the mid-cochlear quarter of a turn...". Please define "cent" here.

Reviewer #2 (Remarks to the Author):

The focus of this manuscript from Jean et al is to describe the presence of functional coupling between the sensory IHCs of the adult cochlea of mice and gerbils, which was previously unknown. Using a combination of approaches, including dye labelling, whole-cell patch clamp recordings, high-resolution imaging and computational modelling, the authors have described the presence of mini-syncytia among IHCs (between 2 and 9 cells) in post-hearing mice. These syncytia appear to increase from just 2% of IHCs during the third post-natal week to about 30% at 5-6 weeks of age. The different number of IHCs connected within a mini-syncytium was indicated by results including the diffusion of fluorescent dyes among the cells, a correspondingly large cell membrane capacitance and reduction in input resistance. The fact that gap-junction blockers did not close these cellular connections indicates that connexin gap junction channels are unlikely to be involved. The article also provides high-resolution microscopy data showing "putative" fusion sites

among IHCs, and a computational model aimed at showing the potential functional benefit of having such syncytia among IHCs. Overall, this work provides a descriptive investigation for the possible presence of IHC-IHC connections in the adult mammalian cochlea. Although the conclusions appear substantiated by some of the data presented, there are several aspects of the work that need some additional consideration in order to rule out the presence of any potentially confounding scenarios that could affect the interpretation of the data.

1) The description of the work is very superficial with minimal quantification and a complete absence of statistical analysis. Although the word "significance" is mentioned in some places, there is no mention of the P value anywhere in the text nor the specific test that was done. In addition, it is difficult to extrapolate how many experiments have been performed, including the number of biological replicates used for each set of data. For example, was the IHC coupling mainly found in some preparations/animals? This should be clearly described in the ms. This lack of information makes it very difficult to assess the robustness of the data provided. The methods section is also very poorly written, with minimal detail.

2) It is not clear to me why the mice have to be over 1 month old in order to show a substantial increase in the coupling between IHCs. From a purely functional point of view, if this phenomenon is so crucial for proper sound perception, why is it not present during the first month of life? Also, does the IHC-IHC coupling increase further at older ages? If so, then the interpretation is likely to change substantially.

3) Figure 2f is interesting but also highlights some concerns. The level of variability in the size of the calcium current recorded from an "uncoupled" IHCs is very variable (from about 50 pA to 450 pA), which is not consistent with work previously published from this lab. Is it possible that this large variability is due to differences in the general health of the patched IHCs? Is the correlation shown in Fig 2f significantly different from 0, given the large current size variability? It would be useful to show the rest of the IHCs as a function of age – does it become larger?

4) If coupling between IHCs is present, then a similar variability in the outward K current size to that seen for IcaMax would be expected. Considering that an adult IHC showing a Cm of about 9pF and a IK of about 15nA (e.g. from your previous study: Sendin et al., 2007), IK would be expected to be about 130nA if 9 IHCs are connected? You should provide some examples of very large K currents to corroborate the calcium current results. This is important because K currents are less likely to be affected by the physiological state of the cells than the calcium currents.

5) The use of the perforated patch configuration is a crucial part of the present work. Despite this, there is no information regarding how it has been performed (one sentence in the Method section, which does not say very much), and which control experiments have been done to make sure whole-cell was not achieved.

6) The morphological images are very nice, although, as the authors mention in the text, they only provide limited evidence for a "putative" representation of the IHC-IHC fusion sites. It is very difficult to associate the small membrane gaps observed in very few images to the data presented in the first part of the paper. The dimensions of the fusion sites provided in Table 1 were obtained from 7 identified sites, however, there is no indication of how many IHCs/mice have been looked at in total. If 30% of IHCs are connected, then it would be reasonable to expect to find a much larger number of fusion sites. Unfortunately, as mentioned above, there is an underlying possibility that all changes observed are due to experimental artifacts, which are not fully excluded by the authors. Is it possible to obtain high-resolution images from the same IHCs used for dye labelling?

7) Assuming that the number of coupled IHCs does not change further in older animals (see above), the functional explanation for these syncytia is not very clear. Although, there are a lot of speculative hypotheses listed in the manuscript, there is not a clear understanding of the distribution of these "putative" syncytia that occurs in about 30% of IHCs. Their random distribution along the cochlea would be differentially "diluted" with the remaining 70% of IHCs that do not show any functional coupling, making it difficult to really appreciate the functional benefit. Moreover, it is difficult to reconcile why the afferent fibers undergo extensive pruning during development, in order to make one-to-one connections with the IHCs, and then after a couple of

weeks revert back to a configuration resembling that of the immature cochlea.

Reviewer #3 (Remarks to the Author):

The manuscript "Macromolecular and electrical coupling between inner hair cells in the rodent cochlea" by Jean and colleagues (NCOMMS-19-40978) is a very detailed and powerful study of a novel type of communication between cochlear inner hair cells (IHC), namely direct junctions in the membrane of adjacent IHC. The authors demonstrate the presence of these junctions, which couple up to 9 adjacent IHCs, using dye diffusion, patch-clamp electrophysiology and FIB-SEM reconstructions. They then demonstrate potential function of the junctions in hearing by demonstrating that activation of one IHC can cause pre-synaptic calcium increases at ribbons of neighboring coupled IHC using an especially elegant technique of imaging CtBP2 protein conjugated to a calcium sensor, which diffuses between neighboring cells. They conclude with a computational model demonstrating a modest improvement in coordination of IHC activity, with possible benefits for sound encoding. While this work raises many interesting questions about the function of these "mini-syncytia" in hearing for future studies, it is a thorough and well-done manuscript. A few concerns remain:

Major:

- The main concern with this work, which is partially, but not completely, addressed, is that of damage to the tissue during in vitro experiments. While in vivo measurement of this phenomenon is not currently possible, the authors should show clearly the results of their experiments in which techniques minimal tissue damage (supporting cell removal) were used. They are mentioned in the discussion as well as the supplementary methods. However, the results of these experiments were not mentioned (why?). Please move this data to the main results text, preferably with a figure detailing any differences between the two dissection conditions. This is especially important because the authors note an increase in mini-syncytia occurrence and # of IHC involved with age, but the cochlear tissue becomes more fragile with age as well. The authors notes that coupling between IHC and SCs was not observed, which supports their hypothesis, but SCs could have different membrane components, extracellular matrix molecules, etc that could make them more resistant to damage.
- The introduction details other forms of coupling among neurons. However, cochlear hair cells are not neurons, they are epithelial cells. A discussion of similar junctions (or lack thereof) in this cell type is warranted, and may help support the experimental results.
- In table 1, how many cells were imaged and analyzed, including cells with no fusion or filopodial sites?
- The experiments presented in figure 7 are excellent, showing that functional effect of connected IHCs in potentially stimulating each other's pre-synaptic active zones. Is there any difference in the number of spots activated, or the fluorescence intensity of the spots, with distance from the patched IHC? This is not required for the main body of the manuscript, but could be added to the supplement. This data could give an idea of the effect of distance in determining function of the connected cells.

Minor:

- Please add a clock to the supplementary movies
- Page 4, 5th line from bottom (line numbering would help), "sound born" = "sound borne"
- Page 4, 3rd line from bottom: delete "already"
- Figure 2f – define the red 'plus' symbols in the legend

Reviewer #4 (Remarks to the Author):

The article from Jean et al reports that about 30% inner hair cells in rodents appear to form a syncytium. The experimental breadth includes classical neurobiology methods as well as three-dimensional light and electron microscopy.

I am here to judge the quality of the imaging, and technically from my perspective, there are no

problems. The choice to perform TEM tomography is correct as the junctions and the membrane contacts are difficult to demonstrate with lower resolution approaches. The preparation protocol is standard and in my opinion appropriate to the task. maybe the presence of tannic acid is a little excessive given the fine details understudy, but overall I do not see it as a blocking issue.

As an outsider to the field, I question the general interest of the study, which is from my perspective very limited and more appropriate to a neurobiology journal.

Rev#1

This exceptionally interesting manuscript describes the findings that may significantly change current understanding of the cell-to-cell communications in the mammalian organ of Corti. According to the wisdom of the field, the major mechanosensory receptors of the inner ear – the inner hair cells (IHCs) – send their signals to the auditory nerve fibers separately and do not communicate to each other. However, the authors of this study discovered that, at the latest stages of postnatal development, IHCs form functional mini-syncytia interspersed between non-coupled IHCs along the tonotopic axis of the organ of Corti. The existence of these three-to-nine cells mini-syncytia was confirmed by dye diffusion, patch-clamp recordings, calcium imaging, and 3D electron microscopy. The data are of high-quality and quite convincing. Even though the functional significance of the IHC coupling remains purely speculative, the study may have a profound impact on our understanding of the mechanisms of sound encoding in the mammalian cochlea. I was very excited by this story and have only few minor comments.

First, we would like to thank the reviewer for the appreciation of our multidisciplinary study. The constructive evaluation of our work has helped us to further improve the manuscript. We have addressed all concerns raised by the reviewer.

As the authors correctly pointed out, the major concern is a potential artefactual IHC fusion due to cellular damage in an ex vivo preparation. The authors outlined in the Discussion several reasons to believe that such an artefact is unlikely. However, I would be more comfortable if the authors would show, on a separate figure, the data of a crucial experiment that they described as, “IHC recordings revealing Ca²⁺ current amplitudes worth multiples of the single IHC Ca²⁺ current were also observed ... without removing the contacting supporting cells.”

Done, as requested we provide an additional Supplemental figure (new Fig. S4). We now state in page 8:

“Similar observations were made i) from ruptured-patch recordings from IHCs of the excised apical and basal gerbil organ of Corti (P14-38, Fig. S2), ii) from perforated-patch recordings of mouse IHCs in the excised organ of Corti ($n = 14$ IHCs, $N_{animals} = 11$, P15-P36), and iii) from ruptured-patch recording of IHCs when only one inner sulcus cell was removed for access of the pipette in the excised apical mouse organ of Corti ($n = 15$ IHCs; $N_{animals} = 6$, P16-P24, Fig. S4).”

Specific comments:

Figure 2f: Please clarify which values the data points (open circles and red crosses) represent. I assume I_{CaMax} and I_{rest}, correspondingly. If so, why I_{CaMax} data are presented separately for ruptured and perforated-patch experiments, while I_{rest} data are combined? Exactly which data were used for linear fits? There are three color-coded groups and three linear fits could be expected.

Correct: open circles for the I_{CaMax} , crosses for I_{rest} . We had combined the ruptured-patch and perforated-patch data for both I_{CaMax} and I_{rest} and the black and red lines represent fits to the C_m relationships of I_{CaMax} and I_{rest} , respectively. For clarity, and in response to the reviewer's concern, we have modified Fig. 2e now plotting I_{CaMax} and former 2f (now 2e') now plotting I_{rest} of single and dye coupled IHCs (recorded at 5mM $[Ca^{2+}]_e$) against their respective membrane capacitance. The former Fig. 2f combining different conditions has now become Fig. S1e,e', where we reduced the heterogeneity of the recording conditions and only present the ruptured-patch data. The perforated patch-data are now shown in Fig. S4 in separation together with the minimal cleaning approach data. We have now clarified this in the text of the results section as well as in the figure legends.

"As expected, a correlation with C_m was found for I_{CaMax} ($r = -0.66$, $p < 0.00001$, Fig. 2e) as well as for I_{rest} (Fig 2e', $r = -0.74$, $p < 0.00001$). In a larger data set combining different $[Ca^{2+}]_e$ (1.3, 2 or 5 mM), correlations of were also found for I_{CaMax} and I_{rest} with C_m (I_{CaMax} , $r = -0.66$, $p < 0.00001$, I_{rest} , $r = -0.64$, $p < 0.00001$, P15-51, Fig. S1e,e'). Similar observations were made i) from ruptured-patch recordings from IHCs of the excised apical and basal gerbil organ of Corti (P14-38, Fig. S2), ii) from perforated-patch recordings of mouse IHCs in the excised organ of Corti ($n = 14$ IHCs, $N_{animals} = 11$, P15-P36), and iii) from ruptured-patch recording of IHCs when only one inner sulcus cell was removed for access of the pipette in the excised apical mouse organ of Corti ($n = 15$ IHCs; $N_{animals} = 6$, P16-P24, Fig. S4)."

Figure S2c: Please clarify which values the data points (gray open circles and red crosses) represent. I assume I_{CaMax} and I_{rest} , correspondingly. What is the group with $n=49$? Is that a total number of the cells? Why is the number of cells in panel c not matching to the number of cells in panel b?

Correct: open circles for the I_{CaMax} , crosses for I_{rest} ! We have added this information in the figure legend. We would like to thank reviewer for spotting a mistake in numbers. The graphs have been updated and the numbers corrected now. The number of cells in panel c is matching the number of cells in panel b (19 cells with $C_m < 15$ pF, 25 cells with 15 pF $\leq C_m < 30$ pF, and 9 cells with $C_m \geq 30$ pF = 53 cells).

Legends to figure 3, 4, and 7 are missing the ages of the preparations.

We have now made sure the age is specified in results section, figure and legend.

Results, last paragraph: "...we estimate that a single IHC covers approximately 3 cents on the tonotopic map of the mid-cochlear quarter of a turn...". Please define "cent" here.

Done

"For the human cochlea, we estimate that a single IHC covers approximately 3 cents (1 cent is 1 percent of a semitone, one semitone is $1/12^{th}$ of an octave) on the tonotopic map of the mid-cochlear quarter of a turn."

Rev#2

The focus of this manuscript from Jean et al is to describe the presence of functional coupling between the sensory IHCs of the adult cochlea of mice and gerbils, which was previously unknown. Using a combination of approaches, including dye labelling, whole-cell patch clamp recordings, high-resolution imaging and computational modelling, the authors have described the presence of mini-syncytia among IHCs (between 2 and 9 cells) in post-hearing mice. These syncytia appear to increase from just 2% of IHCs during the third post-natal week to about 30% at 5-6 weeks of age. The different number of IHCs connected within a mini-syncytium was indicated by results including the diffusion of fluorescent dyes among the cells, a correspondingly large cell membrane capacitance and reduction in input resistance. The fact that gap-junction blockers did not close these cellular connections indicates that connexin gap junction channels are unlikely to be involved. The article also provides high-resolution microscopy data showing “putative” fusion sites among IHCs, and a computational model aimed at showing the potential functional benefit of having such syncytia among IHCs. Overall, this work provides a descriptive investigation for the possible presence of IHC-IHC connections in the adult mammalian cochlea. Although the conclusions appear substantiated by some of data presented, there are several aspects of the work that need some additional consideration in order to rule out the presence of any potentially confounding scenarios that could affect the interpretation of the data.

First, we would like to thank the reviewer for the appreciation of our multidisciplinary study. The constructive evaluation of our work has helped us to further improve the manuscript. We have addressed all concerns raised by the reviewer.

1) The description of the work is very superficial with minimal quantification and a complete absence of statistical analysis. Although the word “significance” is mentioned in some places, there is no mention of the P value anywhere in the text nor the specific test that was done. In addition, it is difficult to extrapolate how many experiments have been performed, including the number of biological replicates used for each set of data. For example, was the IHC coupling mainly found in some preparations/animals? This should be clearly described in the ms. This lack of information makes it very difficult to assess the robustness of the data provided. The methods section is also very poorly written, with minimal detail.

In brief, the dye and electrical coupling of IHCs was consistently observed throughout the study by multiple experimenters in different labs: P.J., T.A., M.GD.A., D.Ö., J.F.A., in mice from different breeding colonies and vendors over several years as well as in gerbils. The number of analyzed cells was provided throughout and in our view already reflected the robust data base for our report.

In response to the reviewer’s comment, we have now provided more details on how the data was acquired in the different labs. Specifically, we have placed the information on number of observations, cells, animals, statistical tests used at each instance (in addition to the methods section where the information had been summarized).

2) It is not clear to me why the mice have to be over 1 month old in order to show a substantial increase in the coupling between IHCs. From a purely functional point of view, if this phenomenon is so crucial for proper sound perception, why is it not present during the first month of life? Also, does the IHC-IHC coupling increasing further at older ages? If so, then the interpretation is likely to change substantially.

At present we do not have a sensible reply for why biology might employ less IHC coupling for hearing at earlier stages. Mechanistically, it seems that the extent of supporting cell separation declines during postnatal maturation which gives rise to more opportunities for IHCs to contact each other in the more mature state.

In order to address the reviewer's question we have revisited our data on mice and gerbils but did not observe a clear change of the incidence of IHC coupling in the older animals. Since the experiments were not designed to determine the progression of coupling with age in the oldest age group, this sample is not sufficiently large to determine a trend in this population.

Mice:

We use the temporal bone preparation in which the largest sample of recording in older mice was available. The 71 mice in the total sample of temporal bones contained approximately equal proportions, 30 females, 34 males and 7 unspecified. There was no obvious effect of gender. The remaining 29 mice in the coupling histogram were not specified either for age or sex. 15 of the age specified mice were older than P100 but still could be readily recorded in the 2P configuration. Of these, the oldest where 3 cells were found to be coupled was P350. These were C57BL6 animals that were donated and recorded more out of curiosity. At this stage, we have insufficient information to carry out any analysis in a systematic manner and so these data will be excluded here. The plot shows the numbers of loaded cells versus the age for mice with ages less than P100. The data can be replotted to show the age distribution for each set of coupled cells (from 1-8 cells). As can be seen coupling was found at all ages in this sample. Since the experiments were not designed to determine the progression of coupling with age, the sample is not sufficiently large to determine a trend in this population.

Figure 1 of this rebuttal letter: Coupling is found at all ages in mice.

(a) The plot shows the numbers of loaded IHCs versus the age for mice with ages less than P100. (b) The data can be replotted to show the age distribution for each set of labeled IHCs (from 1-8 IHCs). As can be seen coupling was found at all ages in this sample.

Data sample: IHC number in recording, rounded mean postnatal age (P), n of recordings
 1 IHC (non-coupled), P48, $n = 31$; 2 IHCs, P48, $n = 15$; 3 IHCs, P46, $n = 6$; 4 IHCs, P60, $n = 2$;
 5 IHCs, P48, $n = 3$; 6 IHCs, P58, $n = 1$; 8 IHCs, P54, $n = 1$

Gerbil (dissected organ of Corti)

Figure 2 of this rebuttal letter: Coupling is found at all ages in Gerbils.

(a) distribution of the number of IHCs labelled during a dye-loading experiment as a function of age.

(b) I_{rest} of un(non)-coupled IHCs before 28 and from P28 was not statistically different (<P28: -26.79 ± 2.32 pA, $n = 39$, $N=15$; $\geq P28$: -28.16 ± 1.96 pA, $n = 24$, $N=9$, Mann-Whitney-Wilcoxon test, $p = 0.47$)

3) Figure 2f is interesting but also highlights some concerns. The level of variability in the size of the calcium current recorded from an “uncoupled” IHCs is very variable (from about 50 pA to 450 pA), which is not consistent with work previously published from this lab. Is it possible that this large variability is due to differences in the general health of the patched IHCs? Is the correlation shown in Fig 2f significantly different from 0, given the large current size variability? It would be useful to show the I_{rest} of the IHCs as a function of age –does it become larger?

We fear there was a misunderstanding of the reviewer regarding the maximal Ca^{2+} current amplitude: Fig. 2f plots Ca^{2+} current amplitude of non-coupled IHCs and IHCs coupled to various extents (as seen from the different membrane capacitance (C_m)). There is no case of a non-coupled IHC with a maximal Ca^{2+} current amplitude of 450 pA! In fact, even in 5 mM $[Ca^{2+}]_e$ the largest maximal Ca^{2+} current amplitude of a non-coupled IHC was 300 pA. The 4 coupled cells presenting a maximal calcium current of 450 pA for a capacitance of only 10 pF in Fig 2f, was due to an incomplete compensation of the slow capacitance often encountered with large mini-synctia of 3-4 cells.

Moreover, we would like to stress that these currents were recorded in different conditions and we combined them here to illustrate the robustness of the finding of the scaling of Ca^{2+}

current and holding current amplitudes with the extent of coupling. Specifically data of former Fig. 2f were acquired at different different age (postnatal days 14-56) and at extracellular $[Ca^{2+}]_e$ (1.3, 2 or 5 mM) in ruptured or perforated-patch recording configurations in different protocols (maximum current in protocols to establish the current-voltage relation or single step depolarization targeting the potential that elicits the maximal Ca^{2+} influx). This had been indicated in the figure legend and explains the substantial variability. It is despite this variability that we found a correlation of the maximal Ca^{2+} current amplitude with the membrane capacitance ($r = -0.66$, $p < 0.00001$) which further corroborates the notion of low-resistance electrical coupling.

The Ca^{2+} current data also argue strongly against the concern of the reviewer that IHC coupling might reflect a compromised physiological state. As pointed out by the reviewer in the following comment, compromised general health of the coupled cells would be expected to cause lower Ca^{2+} currents. However, instead, the recordings from coupled cells indicate a summation of normally sized Ca^{2+} currents in the limits of the expected poorer voltage clamp. Therefore, by the reviewer's argument #4, the fact that the Ca^{2+} current scales with the number of coupled IHCs in the mini-syncytium supports the good general health of the recorded IHCs, which is the standard in our recordings. For clarity, and in response to the reviewer's concern, we have modified Fig. 2e now plotting I_{CaMax} and former 2f (now 2e') now plotting I_{rest} of single and dye coupled IHCs against their respective membrane capacitance at 5mM $[Ca^{2+}]_e$. The former Fig. 2f combining different conditions has now become Fig. S1e,e', where we reduced the heterogeneity of the recording conditions and only present the ruptured-patch data. The single non-coupled IHC data is taken from a previous paper from the Göttingen lab (Jean et al., 2018) recorded by the same experimenter, using the same protocol, where the resting currents never exceeded -50 pA (mean of -22 pA). The perforated patch-data are now shown in Fig. S3 together with further ruptured-patch data obtained with the minimal cleaning approach.

In response to the reviewers suggestion to look at I_{rest} as a function of age, we refer to the Figure 2b of this rebuttal letter which shows based on gerbil recordings that there was no significant difference between recordings from IHCs before and from P28.

4) If coupling between IHCs is present, then a similar variability in the outward K current size to that seen for I_{CaMax} would be expected. Considering that an adult IHC showing a C_m of about 9pF and a I_K of about 15nA (e.g. from your previous study: Sendin et al., 2007), I_K would be expected to be about 130nA if 9 IHCs are connected? You should provide some examples of very large K currents to corroborate the calcium current results. This is important because K currents are less likely to be affected by the physiological state of the cells than the calcium currents.

We hope the reviewer will agree that the large size of the K^+ currents (as indicated by the reviewer) together with the series resistance ($\sim 5-15$ MOhm) and the coupling resistance ($\sim 10-100$ Mohm) would make it impossible to perform the proposed experiment. While the Ca^{2+} currents (hundreds of picoamperes) of the IHCs within a mini-syncytium could typically be recorded reasonably well, this would not be the case for the K^+ currents, which as the reviewer points out, are of nanoampere amplitude and can barely be clamped in the patched cell, let alone in the coupled cells.

In response to the reviewer's concern, we analyzed the membrane conductance as a function of the number of OGB5N-loaded IHCs in the temporal bone preparation. Using Cs^+ to minimize the conductance of K^+ channel we depolarized the patch-clamped IHCs to various levels (100 ms long depolarizations in 10 mV increments from a holding potential of -60 mV (Fig. S5)). The Cs^+ currents increased with the number of dye-filled IHCs, indicating a partial summation over the currents of the coupled IHCs. We note that the recordings underestimate the true currents due to the voltage drop over the pipette series resistance to the patched IHC and due to the junctional resistance among the coupled cells. In order to quantify this result, we analyzed the membrane slope conductance (g_m) of IHCs at -60 and +20 mV. There was a significant positive correlation of g_m and the number of coupled cells was observed at both potentials (-60 mV: $r = 0.78$; +20 mV: $r = 0.77$, both regression slopes significantly different from 0 at $p < 0.05$; Fig. S6). This corroborates our conclusion electrical IHC coupling.

Figure S5. Cesium outward currents increase with the number of dye-coupled IHCs.

Analyzing dye and electrical IHC coupling in temporal bone preparation, using 2PLSM imaging of OGB5N-loaded IHCs (top panel) and IHC recordings of Cs^+ -mediated outward currents (middle panel) in response to 100 ms long depolarizations (in 10 mV increments, bottom panel) from a holding potential of -60 mV. As shown for a sample of $n = 56$ distinct experiments and quantified in Fig. S6, the maximal Cs^+ current increases with the number of dye-filled IHCs, indicating a partial summation over the currents of the coupled IHCs. We note that the recordings underestimate the true currents due to voltage drops over the series resistance to the patched IHC and the junctional resistance among the coupled cells.

Figure S6. IHC membrane conductances increase with the number of dye-coupled IHCs. Membrane slope conductances (G_m) at the holding potential -60 mV and at $+20$ mV from I-Vs for positively identified uncoupled and coupled cells. All the data was collected within 60 s of whole cell break-in with Cs^+ as the major pipette cation. The membrane slope conductance at -60 mV, increasing with the number of coupled cells, is a good estimator of the effect of coupling. In this sample, using internal Cs^+ , $G_m = 2.05 \pm 1.25$ nS (mean \pm SD, $n = 25$ uncoupled cells) at -60 mV. Preliminary data using K^+ as the major pipette cation did not permit the sufficient voltage clamp control to estimate G_m at $+20$ mV. Dashed lines show regression fits, with slopes significantly different from 0 nS/cell ($p < 0.05$).

5) The use of the perforated patch configuration is a crucial part of the present work. Despite this, there is no information regarding how it has been performed (one sentence in the Method section, which does not say very much), and which control experiments have been done to make sure whole-cell was not achieved.

Since we have reported these experiments in our first IHC study 20 years ago, the Göttingen laboratory has used the perforated patch-recordings for the reason of their less invasive and more stable nature. An accidental transition from perforated patch to ruptured patch is very rare for experienced patch-clampers as those contributing to the study and obvious as an instantaneous reduction in the series resistance that is regularly monitored by the membrane capacitance compensation circuitry of the EPC amplifiers used here. This was not observed in any of the recordings of this study.

In response to the reviewer's concern, we now offer a more extensive description here and in the methods section.

"For perforated-patch, the intracellular solution contained (in mM): 130 Cs-methanesulfonate, 10 TEA-Cl, 10 4-AP, 10 HEPES, and 1 MgCl_2 . Amphotericin B (250 $\mu\text{g}/\text{ml}$ Calbiochem, La Jolla, CA) was added from a 1000x stock in DMSO (final DMSO concentration not exceeding 0.1%) during sonication. Pipettes were tip-dipped in amphotericin-free pipette solution for 20-40 sec and then brought to the IHC within less than 1 min to avoid accidental spill of amphotericin which blocks seal formation. Perforation of the patch

membrane usually occurred within 5 min and recordings were started when the series resistance was lower than 30 MOhm. A sudden drop of series resistance, which would be indicative of a spontaneous rupturing of the patch membrane was not observed.”

6) The morphological images are very nice, although, as the authors mention in the text, they only provide limited evidence for a “putative” representation of the IHC-IHC fusion sites. It is very difficult to associate the small membrane gaps observed in very few images to the data presented in the first part of the paper. The dimensions of the fusion sites provided in Table 1 were obtained from 7 identified sites, however, there is no indication of how many IHCs/mice have been looked at in total. If 30% of IHCs are connected, then it would be reasonable to expect to find a much larger number of fusion sites. Unfortunately, as mentioned above, there is an underlying possibility that all changes observed are due to experimental artifacts, which are not fully excluded by the authors. Is it possible to obtain high-resolution images from the same IHCs used for dye labelling?

Using FIB-SEM, we investigated the membranes of 3 IHCs to their respective neighboring IHCs (neighbors were not entered into the IHC count) right after the onset of hearing (2 IHCs from a P15 organ of Corti and 1 IHC from a P16 organ of Corti, $n = 3$ FIB-SEM runs, $N_{animals} = 2$) and 3 IHCs after full maturation (2 IHCs from a P34 organ of Corti and 1 IHC from a P37 organ of Corti, $n = 3$ FIB-SEM runs, $N_{animals} = 2$). This corresponds to a total of 6 FIB-SEM runs. We took this effort, because FIB-SEM is the only method that enables reconstruction of a large volume with close to isotropic and sufficiently small voxel size (3 nm in x/y and 5 nm in z) for faithful inspection of the IHC membrane. We see that the Table 1 and S1 figure were not completely clear on the IHC numbers. We investigated in each run 1 full IHC that makes contact to the two neighboring cells. The neighboring cells, however, are not fully included in the 3D volume and therefore not completely analyzed. We clarified this confusing statement now in the text:

“To identify a structural basis for the mechanism of IHC coupling, we employed 3D electron microscopy (focused ion beam–scanning electron microscopy (FIB-SEM): voxel size: 3 nm in x/y and 5 nm in z). We focused our imaging on the perinuclear basolateral membranes of mouse IHCs that appeared to be in contact in more mature preparations (Figs. 5 and 6; Fig. S8; Movie S4). We investigated the membranes of 3 IHCs to their respective neighboring IHCs (neighbors not being included into the IHC count) right after the onset of hearing (2 IHCs from a P15 organ of Corti and 1 IHC from a P16 organ of Corti, $n = 3$ FIB-SEM runs, $N_{animals} = 2$) and 3 IHCs after full maturation (2 IHCs from a P34 organ of Corti and 1 IHC from a P37 organ of Corti, $n = 3$ FIB-SEM runs, $N_{animals} = 2$).”

Further, we have now prominently included all that information into the results section and specify N to state $N_{animals}$.

We have used the electron microscopy to provide structural insight into the cell biological mechanism underlying the IHC-coupling. While gap junctions were not observed in the basolateral region, we found 8 events with morphological attributes of putative IHC-IHC fusion sites.

The connection between our electrophysiological estimates of contact (12 to 75 MOhm coupling resistance, equivalent to 13 to 80 nS coupling conductance) are in line with the EM

results. The dimensions of the putative intercellular bridges can be converted to resistances assuming a standard intracellular resistance of 150 Ohm · cm. For the bridges listed in table S1 (excluding the outlier), the following tables lists the estimated individual resistances and conductances, as well as the resulting values for all the bridges between two connected cells. Those combined resistances are the inverse of the summed conductances, because they act in parallel to each other.

FIB Run	cells	diameter (nm)	length (nm)	Resistance (MOhm)	Conductance (nS)
P16 #3	IHC2-IHC3	135	35	3.7	272.6
P34 #2	IHC2-IHC3	30	102	216.5	4.6
		35	147	229.2	4.4
		5	124	9472.9	0.1
				110	9.1
P37 #3	IHC3-IHC4	30	40	84.9	11.8
		14.4	15	138.2	7.2
		15.23	15	123.5	8.1
		52.8	55	37.7	26.5
				19	53.7

The resulting resistances between neighboring cells are close to the range estimated with electrophysiology, if you account for an additional resistance of several MOhm across the cells from pipette to pipette.

We think that this counters the reviewer’s argument that the coupling should be reflected in a “much larger number of fusion sites”.

7) Assuming that the number of coupled IHCs does not change further in older animals (see above), the functional explanation for these syncytia is not very clear. Although, there are a lot of speculative hypotheses listed in the manuscript, there is not a clear understanding of the distribution of these “putative” syncytia that occurs in about 30% of IHCs. Their random distribution along the cochlea would be differentially “diluted” with the remaining 70% of IHCs that do not show any functional coupling, making it difficult to really appreciate the functional benefit. Moreover, it is difficult to reconcile why the afferent fibers undergo extensive pruning during development, in order to make one-to-one connections with the IHCs, and then after a couple of weeks revert back to a configuration resembling that of the immature cochlea.

This, to our knowledge is the first report of an unprecedented phenomenon. The finding is robust across various mouse breeds, recording conditions, several experimenters in two labs and generalizes to another rodent species. We have carefully discussed the potential functional implications of IHC-coupling. Future work will be required to further elucidate mechanisms, prevalence and functional role in mice and other species.

Rev#3

The manuscript “Macromolecular and electrical coupling between inner hair cells in the rodent cochlea” by Jean and colleagues (NCOMMS-19-40978) is a very detailed and powerful study of a novel type of communication between cochlear inner hair cells (IHC), namely direct junctions in the membrane of adjacent IHC. The authors demonstrate the presence of these junctions, which couple up to 9 adjacent IHCs, using dye diffusion, patch-clamp electrophysiology and FIB-SEM reconstructions. They then demonstrate potential function of the junctions in hearing by demonstrating that activation of one IHC can cause pre-synaptic calcium increases at ribbons of neighboring coupled IHC using an especially elegant technique of imaging CtBP2 protein conjugated to a calcium sensor, which diffuses between neighboring cells. They conclude with a computational model demonstrating a modest improvement in coordination of IHC activity, with possible benefits for sound encoding. While this work raises many interesting questions about the function of these “mini-syncytia” in hearing for future studies, it is a thorough and well-done manuscript. A few concerns remain:

First, we would like to thank the reviewer for the appreciation of our multidisciplinary study. The constructive evaluation of our work has helped us to further improve the manuscript. We have addressed all concerns raised by the reviewer.

1. Major:

The main concern with this work, which is partially, but not completely, addressed, is that of damage to the tissue during *in vitro* experiments. While *in vivo* measurement of this phenomenon is not currently possible, the authors should show clearly the results of their experiments in which techniques minimal tissue damage (supporting cell removal) were used. They are mentioned in the discussion as well as the supplementary methods. However, the results of these experiments were not mentioned (why?). Please move this data to the main results text, preferably with a figure detailing any differences between the two dissection conditions. This is especially important because the authors note an increase in mini-syncytia occurrence and # of IHC involved with age, but the cochlear tissue becomes more fragile with age as well. The authors notes that coupling between IHC and SCs was not observed, which supports their hypothesis, but SCs could have different membrane components, extracellular matrix molecules, etc that could make them more resistant to damage.

We are well aware of the potential caveats of the explanted organ of Corti preparation and clearly, the submittal reflects our longer-standing effort to move to more mature preparations in order to elucidate the physiology of mature inner hair cells and to better relate the *ex vivo* synaptic physiology to the *in vivo* recordings e.g. from auditory nerve fibers. In the present study, the data were collected by experienced hair cell patch-clampers who applied high quality criteria to their preparations: e.g. DIC imaging showing well preserved hair bundles, lack of membrane blebs (apically or basolaterally), maintained shape/size of IHC and lack of obvious “Brownian motion” inside the IHC. In a subset of experiments, minimal cleaning of supporting cells was used and comparable results were obtained (Fig. S4).

In order to further address the concern of the reviewer we have i) compared parameters that are sensitive to IHC viability and ii) included experiments in a more intact preparation of the cochlea.

i) Findings that speak to the viability of the coupled cells are that the Ca^{2+} current and the resting current of couples of two did not exceed double that of non-coupled cells (1 IHC: $I_{\text{CaMax}} = -161 \pm 15$ pA, $I_{\text{rest}} = -22.05 \pm 2.23$ pA, $n = 21$; 2 IHCs: $I_{\text{CaMax}} = -241 \pm 28$ pA, $I_{\text{rest}} = -32.15 \pm 4.39$ pA, $n = 13$, Fig. 2e). Lower viability of coupled cells would have likely produced more leak (greater resting current) and the poorer condition would have reduced the Ca^{2+} current which is very sensitive (e.g. ref. 1). The resting currents were not significantly different between non-coupled IHCs of $<P28$ and $\geq P28$ gerbils ($<P28$: -26.79 ± 2.32 pA, $n = 39$ IHCs, $N_{\text{animals}} = 15$; $\geq P28$: -28.16 ± 1.96 pA, $n = 24$, $N_{\text{animals}} = 9$, Mann-Whitney-Wilcoxon test, $p = 0.47$). These findings suggest that the developmentally increased probability of coupling does not reflect artificial IHC-IHC fusion due to poorer conditions of the preparation. Instead, we propose that this developmental increase of IHC-coupling results from the extended membrane-membrane contact area due to the reduced separation of IHCs by the supporting cells.

ii) Coupling was also found in the least invasive IHC recording approach in the dissected mouse organ of Corti when only one inner sulcus cell was removed for access of the pipette in the excised apical mouse organ of Corti and IHC membranes were approached with applying positive pressure to the patch pipette ($n = 15$ IHCs; $N = 6$ animals, P16-P24, Fig. S4).

iii) An experimental cochlea preparation with least mechanical alteration of the cochlea was contributed by the Ashmore lab, which worked in an *in situ* preparation of the temporal bone. The temporal bone containing the semicircular canals and the cochlea was removed from the skull and placed in a perilymph like extracellular solution. The bone covering the organ of Corti was carefully opened and Reissner's membrane was removed to gain access to the scala media, exposing a small opening to observe the structures. Patch-clamp of IHCs avoided removal of supporting cells from all but the cells being recorded as no other access to 'clean' membrane for patching was possible. In this set of experiments the same findings were obtained: dye- and electrical coupling, activation of presynaptic Ca^{2+} influx at the active zones of all IHCs of the syncytium, upon depolarization.

We also took all measures possible to exclude the possibility of histological artifacts in EM experiments: dissections were performed by an experienced researcher and we further carefully selected the samples to fulfil our requirements for high quality embeddings and excluded beforehand samples that did not meet our criteria. Further, three researchers independently analysed the data sets and the listed putative fusion sites are the result of this independent analysis. Conventional, fixative based embeddings of the organ of Corti are established in the Wichmann laboratory for many years. Regarding the artefacts of this specific high-contrast embedding, in our view, the 3 criteria employed take additional care of selecting the putative fusion sites and distinguish them from non-traceable membranes that could be present due to the embedding for FIB-SEM: Fusion sites were defined when at least 2 out of 3 criteria were fulfilled: i) a lack of traceable opposing IHC-IHC membranes, ii) seamless transition of the membranes of both IHCs, forming iii) a cytoplasmic bridge. The notion of cytoplasmic bridges was in some cases further supported by the presence of cellular structures such as ribosomes. Therefore, we are confident that the flat cell-cell

contacts are not an artifact due to a specific embedding method. Moreover, flat contacts in older animals were also observed upon high-pressure freezing followed by freeze substitution and subsequent electron tomography, which preserves the tissue in a near-native state. This way, we can exclude that some kind of collapse or shrinkage took place causing the flat cell-cell apposition, which appears in older animals and forms the prerequisite for fusion sites. However, this method is not suitable to check for membrane-to-membrane fusion sites in a systematic manner, because 250 nm semithin-sections need to be performed for tomography and therefore only small regions can be covered.

2. The introduction details other forms of coupling among neurons. However, cochlear hair cells are not neurons, they are epithelial cells. A discussion of similar junctions (or lack thereof) in this cell type is warranted, and may help support the experimental results.

done

3. In table 1, how many cells were imaged and analyzed, including cells with no fusion or filopodial sites?

This information was contained in Table S1 and has now been moved to results section and table 1.

4. The experiments presented in figure 7 are excellent, showing that functional effect of connected IHCs in potentially stimulating each other's pre-synaptic active zones. Is there any difference in the number of spots activated, or the fluorescence intensity of the spots, with distance from the patched IHC? This is not required for the main body of the manuscript, but could be added to the supplement. This data could give an idea of the effect of distance in determining function of the connected cells.

The fluorescence intensity of the spots with distance to the patched IHC is already depicted by Fig 7c. Indeed, this panel shows the averaged spot intensity in C1 (patched IHC), C2 (adjacent IHCs), C3 (two cells away), and C4 (three cells away). However, the number of spots is insufficient to answer whether there is a distance dependent activation, which is a very interesting point. In response to the reviewer's comment we have now compared the amplitude and voltage-dependence of activation of the spots in C1 and C2 (left and right inset of Figure 7c, respectively), and no significant differences were found. We have added the following sentence in the manuscript:

"We did not find differences of the synaptic Ca^{2+} -signals between the patched IHC (C1) and its coupled neighbor(s) (C2, the number of spots analyzed from further distant coupled cells did not permit statistical comparisons). Neither their maximal amplitude ($\Delta F_{\text{max}}/F_0$: 1.48 ± 0.25 , $n = 37$ spots in C1 of 8 mini-syncytia, $N_{\text{animals}} = 8$ vs. 1.88 ± 0.28 , $n = 43$ spots in C2 of 8 mini-syncytia, $N_{\text{animals}} = 8$; $p = 0.31$, Mann-Whitney-Wilcoxon test) nor their voltage-dependence of activation (V_h : -23.66 ± 2.24 mV, $n = 17$ spots in C1 in 7 mini-syncytia, $N_{\text{animals}} = 7$ vs. -23.79 ± 1.59 mV, $n = 30$ spots in C2 in 8 mini-syncytia, $N_{\text{animals}} = 8$; $p = 0.96$, Student t-test; note that because of signal-to-noise requirements the analysis of voltage-dependence could only be performed on a subset of synapses) were different."

Regarding the number of spots activated, a full z-scan of the mini-syncytia would have been required and would have taken several minutes since the cells are not fully aligned in z. Given the difficulty of recording good quality Ca^{2+} signals in such syncytia, the protocol was engaged as fast as possible after breaking in where the most ribbons were observed in all cells.

A supplemental figure S10 has now been added, where we can appreciate more examples of synaptic Ca^{2+} signals in IHC syncytia at different z-coordinates from the data analyzed in Fig. 7. A supplemental Movie S7 has also been added where we see the appearance of Ca^{2+} signals upon depolarization in two coupled IHCs.

5.Minor:

Please add a clock to the supplementary movies

Done for Supplemental Movie 1 and new Supplemental Movie 7. Moreover, scale bars were added as well where appropriate.

- Page 4, 5th line from bottom (line numbering would help), “sound born” = “sound borne”

Done

- Page 4, 3rd line from bottom: delete “already”

Done

- Figure 2f – define the red ‘plus’ symbols in the legend

The figure has been changed for more clarity

Reviewer #4 (Remarks to the Author):

The article from Jean et al reports that about 30% inner hair cells in rodents appear to form a syncytium. The experimental breadth includes classical neurobiology methods as well as three-dimensional light and electron microscopy.

I am here to judge the quality of the imaging, and technically from my perspective, there are no problems. The choice to perform TEM tomography is correct as the junctions and the membrane contacts are difficult to demonstrate with lower resolution approaches. The preparation protocol is standard and in my opinion appropriate to the task. maybe the presence of tannic acid is a little excessive given the fine details understudy, but overall I do not see it as a blocking issue.

First, we would like to thank the reviewer for the appreciation of our study. We used tannic acid for our high-pressure frozen samples because in our experience it gives a superb contrast for our tissue to visualize all required structures. So far, we did not face any problems with this staining in combination with subsequent osmium application.

Peer Review File

Reviewers' comments second round:

Reviewer #2 (Remarks to the Author):

The revised MS has addressed some of my previous concerns, although I am still puzzled as to whether this new morphological observation, which only affects around 30% of IHCs, has any functional role in hearing. What if this observation is part of the slow and progressive degeneration of the aging cochlea? This could be an important point, but this is not even considered by the Authors.

As mentioned in my previous report, is difficult to reconcile why the afferent synapses undergo such extensive pruning and tonotopic refinement during development, only then to revert back to a configuration resembling that of the immature cochlea, a few weeks later. The random distribution of IHC interconnections along the cochlea would be dispersed within the remaining 70% of IHCs that do not show any functional coupling, diluting any functional role these could have. This makes it very difficult to appreciate any functional benefit these IHC connections could have.

It is clear that the Authors have done a large amount of work, and it is very commendable, but in its current format the ms describes a morphological phenomenon with unknown functional implications – and as such is unlikely to be of interest to a broad readership.

Considerations regarding my previous comments:

- 1) More details are now included in the text and Method section.
- 2) This point remains largely unanswered. From a physiological perspective, it does not make sense why such a crucial mechanism (as described by the authors) is absent during the first month of an adult mouse. The hearing ability of a mouse is fully developed well before postnatal day 30. So, why is this phenomenon only observed at later stages? Which sound encoding characteristics are improved or reduced by the appearance of this mechanism. From a general perspective, IHC coupling could also be interpreted as a cochlear dysfunction in a slowing aging mouse. This point is vital for the overall interpretation of the paper.
- 3) This has not been clarified in the revised ms.
- 4) This is another critical point that has not been fully addressed. The reason for not recording the K current is not justified. The authors could easily prove this concept by stepping the membrane potential to moderately depolarizing voltages (for example -40 mV). The data shown using Cs⁺ are impossible to judge due to the fact that the traces are compressed together.
- 5) Thank you for dealing with this point.
- 6) The additional explanation is very useful for interpreting the results.
- 7) This aspect has not been addressed at all in the revised ms. Without any clear understanding of the physiological and functional implications of the findings (see also point 2 above), the ms represents a very descriptive/observational study of an unknown morphological phenomenon, which could have no implication at all in hearing. There is also the possibility, which is not discussed by the authors, that this unusual observation could be linked to cochlear dysfunction (see general comment above). As such, the impact of the work is reduced and does not substantially advance our current understanding of cochlear function or sound processing.

Reviewer #3 (Remarks to the Author):

The authors have addressed my concerns.

Rev#2

The revised MS has addressed some of my previous concerns, although I am still puzzled as to whether this new morphological observation, which only affects around 30% of IHCs, has any functional role in hearing. What if this observation is part of the slow and progressive degeneration of the aging cochlea? This could be an important point, but this is not even considered by the Authors. As mentioned in my previous report, it is difficult to reconcile why the afferent synapses undergo such extensive pruning and tonotopic refinement during development, only then to revert back to a configuration resembling that of the immature cochlea, a few weeks later. The random distribution of IHC interconnections along the cochlea would be dispersed within the remaining 70% of IHCs that do not show any functional coupling, diluting any functional role these could have. This makes it very difficult to appreciate any functional benefit these IHC connections could have. It is clear that the Authors have done a large amount of work, and it is very commendable, but in its current format the ms describes a morphological phenomenon with unknown functional implications – and as such is unlikely to be of interest to a broad readership.

We would like to thank the reviewer for the continued interest in our study.

Considerations regarding my previous comments:

1) More details are now included in the text and Method section. We appreciate that this concern was appropriately addressed.

2) This point remains largely unanswered. From a physiological perspective, it does not make sense why such a crucial mechanism (as described by the authors) is absent during the first month of an adult mouse. The hearing ability of a mouse is fully developed well before postnatal day 30. So, why is this phenomenon only observed at later stages? Which sound encoding characteristics are improved or reduced by the appearance of this mechanism. From a general perspective, IHC coupling could also be interpreted as a cochlear dysfunction in a slowing aging mouse. This point is vital for the overall interpretation of the paper.

quoting previous point 2: “It is not clear to me why the mice have to be over 1 month old in order to show a substantial increase in the coupling between IHCs. From a purely functional point of view, if this phenomenon is so crucial for proper sound perception, why is it not present during the first month of life? Also, does the IHC-IHC coupling increase further at older ages? If so, then the interpretation is likely to change substantially.”

At present we do not have a sensible reply for why biology might employ less IHC coupling for hearing at earlier stages. Mechanistically, it seems that the extent of supporting cell separation declines during postnatal maturation which gives rise to more opportunities for IHCs to contact each other in the more mature state. One might also speculate that immature IHCs experience a different means of synchronization which is mediated by supporting cells releasing ATP and/or K^+ ¹⁻³ thought to functionally group neighboring hair cells and their postsynaptic Spiral Ganglion Neurons (SGNs) for refinement of tonotopic organization throughout the pathway.

Finally, we would like to refer the reviewer to another means of cochlear information mixing that results from the electron microscopy demonstration of a branching of the SGN dendrites in 14% of the SGNs. These SGNs then receive input from more than one active zone of one IHC or even more often two IHCs⁴. SGN dendritic branching has also been described by Nadol and colleagues for the human cochlea⁵. Sampling from several active zones and hair cells is well established for the acoustic papilla in birds and reptiles^{6,7}.

3) This has not been clarified in the revised ms.

quoting previous point 3: "Figure 2f is interesting but also highlights some concerns. The level of variability in the size of the calcium current recorded from an "uncoupled" IHCs is very variable (from about 50 pA to 450 pA), which is not consistent with work previously published from this lab. Is it possible that this large variability is due to differences in the general health of the patched IHCs? Is the correlation shown in Fig 2f significantly different from 0, given the large current size variability? It would be useful to show the I_{rest} of the IHCs as a function of age –does it become larger?"

We are not sure that we understand this comment: we believe we have implemented this in the MS: we separated the recordings and now show them as Fig. 2e and Figs. S1e and S2.

"Macromolecular coupling is accompanied by low resistance electrical coupling of IHCs. Indeed, the maximal voltage-gated Ca^{2+} -influx (I_{CaMax}) and resting currents (I_{rest}) increase with the number of dye-filled IHCs (Fig. 2d,e,e', Fig. S2b). Both parameters were significantly larger for mini-syncytia of 3 and 4 dye-coupled IHCs than for single IHCs (1 IHC: $I_{CaMax} = -161 \pm 15$ pA, $I_{rest} = -22.05 \pm 2.23$ pA, $n = 21$ (data taken from Jean *et al.*, 2018 where similar recording conditions were used); 2 IHCs: $I_{CaMax} = -241 \pm 28$ pA, $I_{rest} = -32.15 \pm 4.39$ pA, $n = 13$; 3 IHCs: $I_{CaMax} = -356 \pm 67$ pA, $I_{rest} = -56.25 \pm 10.28$ pA, $n = 4$; 4 IHCs: $I_{CaMax} = -482 \pm 68$ pA, $I_{rest} = -50.75 \pm 6.99$ pA, $n = 4$ recordings; mean \pm S.E.M, Dunn-Holland-Wolfe non-parametric multiple comparison test, $p < 0.05$ for 3 IHCs and 4 IHCs vs. 1 IHC). Interestingly, the strength of IHC coupling varied as also indicated by the variable spread of the TAMRA peptide (Fig. S1c,c'). Strongly dye-coupled IHCs, showing a rapid dye-spread, exhibited fast monophasic I_{Ca} activation kinetics. In the rare case of weakly dye-coupled IHCs, showing slow dye-spread, I_{Ca} exhibited a multiphasic activation, probably due to poorer voltage-clamp of the coupled IHCs via a larger intercellular resistance (Fig. S1d,d' and Fig. S2b). Likewise, membrane capacitance (C_m) estimates, derived from capacitive currents to voltage steps, increased with the number of dye-coupled IHCs again indicating low resistance connections in the majority of the mini-syncytia. As expected, a correlation with C_m was found for I_{CaMax} ($r = -0.66$, $p < 0.00001$, Fig. 2e) as well as for I_{rest} (Fig 2e', $r = -0.74$, $p < 0.00001$). In a larger data set combining different $[Ca^{2+}]_e$ (1.3, 2 or 5 mM), correlations of were also found for I_{CaMax} and I_{rest} with C_m (I_{CaMax} , $r = -0.66$, $p < 0.00001$, I_{rest} , $r = -0.64$, $p < 0.00001$, P15-51, Fig. S1e,e'). Similar observations were made i) from ruptured-patch recordings from IHCs of the excised apical and basal gerbil organ of Corti (P14-38, Fig. S2), ii) from perforated-patch recordings of mouse IHCs in the excised organ of Corti ($n = 14$ IHCs, $N_{animals} = 11$, P15-P36), and iii) from ruptured-patch recording of IHCs when only one inner sulcus cell was removed for access of

the pipette in the excised apical mouse organ of Corti ($n = 15$ IHCs; $N_{animals} = 6$, P16-P24, Fig. S4).”

4) This is another critical point that has not been fully addressed. The reason for not recording the K current is not justified. The authors could easily prove this concept by stepping the membrane potential to moderately depolarizing voltages (for example -40 mV). The data shown using Cs⁺ are impossible to judge due to the fact that the traces are compressed together.

quoting previous point 4: “If coupling between IHCs is present, then a similar variability in the outward K current size to that seen for IcaMax would be expected. Considering that an adult IHC showing a Cm of about 9pF and a IK of about 15nA (e.g. from your previous study: Sendin et al., 2007), IK would be expected to be about 130nA if 9 IHCs are connected? You should provide some examples of very large K currents to corroborate the calcium current results. This is important because K currents are less likely to be affected by the physiological state of the cells than the calcium currents.”

We are not sure that we understand this comment: We have presented the Cs⁺ currents in the supplement and the slope conductance estimates (Fig. S5 and S6) clearly support the notion of an increase of membrane conductance as a function of the number of IHCs in the syncytium and stated in the rebuttal: “Preliminary data using K⁺ as the major pipette cation did not permit the sufficient voltage clamp control....”

5) Thank you for dealing with this point.

We appreciate that this concern was appropriately addressed.

6) The additional explanation is very useful for interpreting the results.

We appreciate that this concern was appropriately addressed.

7) This aspect has not been addressed at all in the revised ms. Without any clear understanding of the physiological and functional implications of the findings (see also point 2 above), the ms represents a very descriptive/observational study of an unknown morphological phenomenon, which could have no implication at all in hearing. There is also the possibility, which is not discussed by the authors, that this unusual observation could be linked to cochlear dysfunction (see general comment above). As such, the impact of the work is reduced and does not substantially advance our current understanding of cochlear function or sound processing.

quoting previous point 4: “Assuming that the number of coupled IHCs does not change further in older animals (see above), the functional explanation for these syncytia is not very clear. Although, there are a lot of speculative hypotheses listed in the manuscript, there is not a clear understanding of the distribution of these “putative” syncytia that occurs in about 30% of IHCs. Their random distribution along the cochlea would be differentially “diluted” with the remaining 70% of IHCs that do not show any functional coupling, making it

difficult to really appreciate the functional benefit. Moreover, it is difficult to reconcile why the afferent fibers undergo extensive pruning during development, in order to make one-to-one connections with the IHCs, and then after a couple of weeks revert back to a configuration resembling that of the immature cochlea.”

We would like to refer the reviewer to another means of cochlear information mixing that results from the electron microscopy demonstration of a branching of the SGN dendrites in 14% of the SGNs. These SGNs then receive input from more than one active zone of one IHC or even more often two IHCs⁴. SGN dendritic branching has also been described by Nadol and colleagues for the human cochlea⁵. Sampling from several active zones and hair cells is well established for the acoustic papilla in birds and reptiles^{6,7}.

Rev#3

The authors have addressed my concerns.

We would like to thank the reviewer for the support of our study.

1. Tritsch, N. X., Yi, E., Gale, J. E., Glowatzki, E. & Bergles, D. E. The origin of spontaneous activity in the developing auditory system. *Nature* **450**, 50–55 (2007).
2. Sendin, G., Bourien, J., Rassendren, F., Puel, J.-L. & Nouvian, R. Spatiotemporal pattern of action potential firing in developing inner hair cells of the mouse cochlea. *Proc. Natl. Acad. Sci. U. S. A.* **111**, 1999–2004 (2014).
3. Johnson, S. L. *et al.* Position-dependent patterning of spontaneous action potentials in immature cochlear inner hair cells. *Nat Neurosci* **14**, 711–717 (2011).
4. Hua, Y. *et al.* Electron Microscopic Reconstruction of Neural Circuitry in the Cochlea. *bioRxiv* 2020.01.04.894816 (2020) doi:10.1101/2020.01.04.894816.
5. Nadol, J. B. Serial section reconstruction of the neural poles of hair cells in the human organ of Corti. I. Inner hair cells. *The Laryngoscope* **93**, 599–614 (1983).
6. Fischer, F. P. Quantitative analysis of the innervation of the chicken basilar papilla. *Hear. Res.* **61**, 167–178 (1992).
7. Sneary, M. G. Auditory receptor of the red-eared turtle: II. Afferent and efferent synapses and innervation patterns. *J. Comp. Neurol.* **276**, 588–606 (1988).